# Accelerated redevelopment of vocal skills is preceded by lasting reorganization of the song motor circuitry

**Michiel Vellema[1,2]\*, Mariana Diales Rocha[1], Sabrina Bascones[3], Sándor Zsebők[4], Jes Dreier[5], Stefan Leitner[1], Annemie Van der Linden[2], Jonathan Brewer[5], Manfred Gahr[1]**

[1]Department of Behavioural Neurobiology, Max Planck Institute for Ornithology, Seewiesen, Germany; [2]Bio Imaging Lab, University of Antwerp, Antwerp, Belgium; [3]Program for Inflammatory and Cardiovascular Disorders, Institut Hospital del Mar d'Investigacions Mèdiques, Barcelona, Spain; [4]Behavioural Ecology Group, Department of Systematic Zoology and Ecology, Eötvös Loránd University, Budapest, Hungary; [5]Department of Biochemistry and Molecular Biology, University of Southern Denmark, Odense, Denmark

**Abstract** Complex motor skills take considerable time and practice to learn. Without continued practice the level of skill performance quickly degrades, posing a problem for the timely utilization of skilled motor behaviors. Here we quantified the recurring development of vocal motor skills and the accompanying changes in synaptic connectivity in the brain of a songbird, while manipulating skill performance by consecutively administrating and withdrawing testosterone. We demonstrate that a songbird with prior singing experience can significantly accelerate the re-acquisition of vocal performance. We further demonstrate that an increase in vocal performance is accompanied by a pronounced synaptic pruning in the forebrain vocal motor area HVC, a reduction that is not reversed when birds stop singing. These results provide evidence that lasting synaptic changes in the motor circuitry are associated with the savings of motor skills, enabling a rapid recovery of motor performance under environmental time constraints.
DOI: https://doi.org/10.7554/eLife.43194.001

**\*For correspondence:**
vellema@gmail.com

**Competing interests:** The authors declare that no competing interests exist.

## Introduction

Complex motor skills, such as singing or playing an instrument, are not inherently determined but need to be acquired through repetitive practice. Many specialized skills are used only incidentally however, and skill performance degrades during intermittent periods of non-use. This means that skills need to be re-acquired each time the need arises. For example, trained human surgeons need to re-acquire their surgical proficiency after a period of absence (*Crewther et al., 2016*), Capuchin monkeys need to acquire manipulative foraging skills to retrieve difficult-to-access, seasonally available food items (*Eadie, 2015*), and songbirds need to re-develop high-quality songs to attract a mate after wintering or long-distance migration (*Slagsvold, 1976*). Whereas learning novel motor skills is generally a lengthy process, the short-term availability of suitable mates and food items imposes a considerable time pressure on the re-acquisition of such specialized skills.

Birdsong, an established model system for vocal motor learning (*Brainard and Doupe, 2013*), is characterized by a high degree of complex and rapid acoustic modulations. The fine motor skills that are necessary to produce such complex, high quality vocalizations are thought to advertise an individual's quality through performance-related characteristics such as song complexity and/or production rate (*Podos, 1997*; *Drăgănoiu et al., 2002*; *Holveck and Riebel, 2007*; *Byers et al., 2010*).

**eLife digest** Developing a complex skill, for example learning how to play the violin, takes considerable time and effort. If you then abandon the violin for months or years, your ability to play will deteriorate over time. However, when you do pick up the violin again you will be able to recover your proficiency much faster than you did when learning for the first time.

It therefore seems that while first learning a complex motor skill, some sort of memory is formed and stored. Learning and memory in general rely on the patterns of connections, called synapses, among neurons in the brain. Early in development, neurons make too many of these connections. This network is then refined over time as unneeded connections are discarded. However, the structural changes to the network that make it easier to re-acquire a skill were not well understood.

Canaries are a useful example of this sort of learning. Young males learn complex songs that are critical for their ability to attract a mate, and can quickly re-acquire their songs at the beginning of each mating season. Female canaries do not normally sing but will develop songs if they are implanted with a testosterone-releasing device. When the implant is removed, they stop singing.

To investigate how songbirds re-acquire songs, Vellema et al. gave female canaries a testosterone implant, and the birds gradually developed songs. The implant was then removed, and for two and a half months the birds did not sing. Vellema et al. then gave the canaries a second testosterone treatment. The canaries rapidly began to sing songs that were strikingly similar to the ones they sang during the initial learning period.

Examining the brains of these canaries revealed that major structural changes in brain connections occurred while the canaries first developed their songs. After the initial period of testosterone exposure, the female canaries had fewer synapses in a brain region that is associated with learning and producing motor tasks. Importantly, this reorganization of the brain circuitry was irreversible.

These findings provide fundamental insights into how we learn and maintain new motor skills. Similar rapid re-learning phenomena are observed in other areas of neuroscience, and future research can explore whether the mechanism described here extends to these areas. Possibly, this mechanism could also illuminate how we can regain skills lost after trauma or injury.
DOI: https://doi.org/10.7554/eLife.43194.002

When juvenile songbirds are one to two months old they start singing noisy, unstructured 'subsongs' akin to human babbling (*Doupe and Kuhl, 1999*), which gradually develop into variable, but recognizable species-specific 'plastic songs' (*Marler and Peters, 1982a*; *Nottebohm et al., 1986*; *Tchernichovski et al., 2001*; *Mori et al., 2018*). With extensive vocal rehearsal the songs slowly consolidate into high performance 'crystallized songs', a process that can take up more than five months in some songbird species (*Nottebohm et al., 1986*). For adult songbirds that annually need to re-acquire their songs this process of juvenile song development greatly exceeds the time that is available to attract a mate during the early breeding season. Thus, adult songbirds must considerably speed-up song re-acquisition to quickly produce songs of adequate quality to impress conspecifics. It is currently unclear how songbirds cope with the seasonally imposed time pressure on song development, and what mechanisms may be involved to ensure reproductive success under such time constraints in nature.

The seasonal development and degradation in vocal performance are highly correlated with changes in testosterone levels (*Nottebohm et al., 1987*; *Smith et al., 1997*; *Tramontin et al., 2000*; *Voigt and Leitner, 2008*), and are often accompanied by gross anatomical and cytoarchitectural restructuring of the brain areas involved in song production (*Nottebohm, 1981*; *Gahr, 1990*; *Kirn et al., 1994*; *Kafitz et al., 1999*; *Thompson and Brenowitz, 2005*; *Balthazart et al., 2008*; *Vellema et al., 2014*). Thus testosterone-regulated adaptations of the songbird brain may play an important role in optimizing song performance. In this study we used adult female canaries (*Serinus canaria*) to investigate how a periodically acquired motor behavior attains a high performance level under developmental time pressure. Although female canaries rarely sing spontaneously, and never produce high quality songs under natural conditions (*Pesch and Guttinger, 1985*), the systemic application of testosterone leads to the development of high-performance song with male-typical

features (*Leonard, 1939*; *Shoemaker, 1939*; *Vallet et al., 1996*). This trait provides us with a well-defined behavioral baseline, and allows us to manipulate in detail the onset and offset of song development through repeated testosterone treatments.

Here we demonstrate that during a protracted testosterone treatment, adult female canaries gradually develop stable, species-typical songs through a process of song crystallization similar to what is known for naturally-raised juvenile male canaries (*Mori et al., 2018*). Re-treatment with testosterone several months after birds have stopped singing, leads to a rapid recurrence of song performance with song features that strongly resemble those after the first treatment. We propose that once developed, vocal motor memories are retained for an extended period of time, enabling adult animals to recover song performance quickly at a later time, a process termed 'savings' (*Ebbinghaus, 2013*). We further demonstrate that neurons in the songbird's premotor nucleus HVC (proper name), a central nucleus in the brain circuitry that controls song production (*Nottebohm et al., 1976*), irreversibly loses a significant number of dendritic spines during the initial testosterone-induced development of vocal skills. This state of reduced spine density is subsequently maintained over long periods in which testosterone levels are low and vocal skills are not used. The observed lasting synaptic pruning could play an important role in the formation of vocal skill savings, enabling birds to rapidly re-acquire song performance within a restricted time period.

## Results

### Testosterone-induced song development in female canaries

To study the development of vocal skills in adult female canaries we recorded and analyzed the entire song ontogeny in acoustically separated animals ($5.6 \pm 0.6$ million song syllables per bird), while manipulating song output by consecutively implanting, removing, and re-implanting birds with testosterone implants.

Whereas no songs were observed prior to treatment, the systemic application of testosterone ($T_1+$) stimulated birds to start singing their first songs after three days ($3.22 \pm 0.46$ days). The phases of female song development were similar to those previously reported for male canaries (*Figure 1*) (*Nottebohm et al., 1986*; *Mori et al., 2018*; *Weichel et al., 1986*). The first subsongs were produced in an irregular fashion and consisted of unstructured syllables with few distinctive features. Songs gradually developed into plastic songs in which different phrases of repeated syllables could be clearly distinguished, albeit with variation between syllables of the same type. During the following six months, female canary songs continued to develop, gradually crystallizing into the species-typical stable songs. Although structurally similar to male canary songs, female songs had relatively small syllable repertoires ($6.2 \pm 1.1$ syllable types), consistent with previous reports (*Vallet et al., 1996*; *Fusani et al., 2003*; *Hartog et al., 2009*).

### Re-application of testosterone triggers an accelerated re-acquisition of song performance

To investigate the birds' abilities to re-acquire song performance after a period without vocal practice we withdrew testosterone ($T_1-$) from the animals, completely abolishing singing behavior in approximately three days ($3.14 \pm 0.63$ days). After 2½ months of no song production birds were treated for a second time with testosterone ($T_2+$), inducing song output after three days ($3.14 \pm 0.74$ days). No subsongs were observed after the second testosterone treatment, and all birds were able to produce plastic songs from the first day of singing.

We first compared the development of temporal song features during the first and second testosterone treatments (*Figure 2*, *Figure 2—figure supplement 1*, *Figure 2—figure supplement 2*). Particularly the speed at which subsequent syllables are repeated, the syllable repetition rate (SR), has been strongly associated with individual performance (*Podos, 1997*; *Podos, 1996*) and has been shown to increase the attractiveness of the song to conspecifics (*Vallet et al., 1998*; *Vallet and Kreutzer, 1995*). We observed that during the first two weeks of testosterone-induced female song development, all syllables were typically produced at a similar rate (SR: $10.3 \pm 1.3$ Hz). While practicing the song, different syllables were gradually sung at different rates, becoming more distinct towards song crystallization (*Figure 2A*). Once crystallized, syllables could be distinguished as fast syllables (SR: $21.7 \pm 1.0$ Hz), medium-speed syllables (SR: $11.9 \pm 1.1$ Hz), and slow syllables (SR:

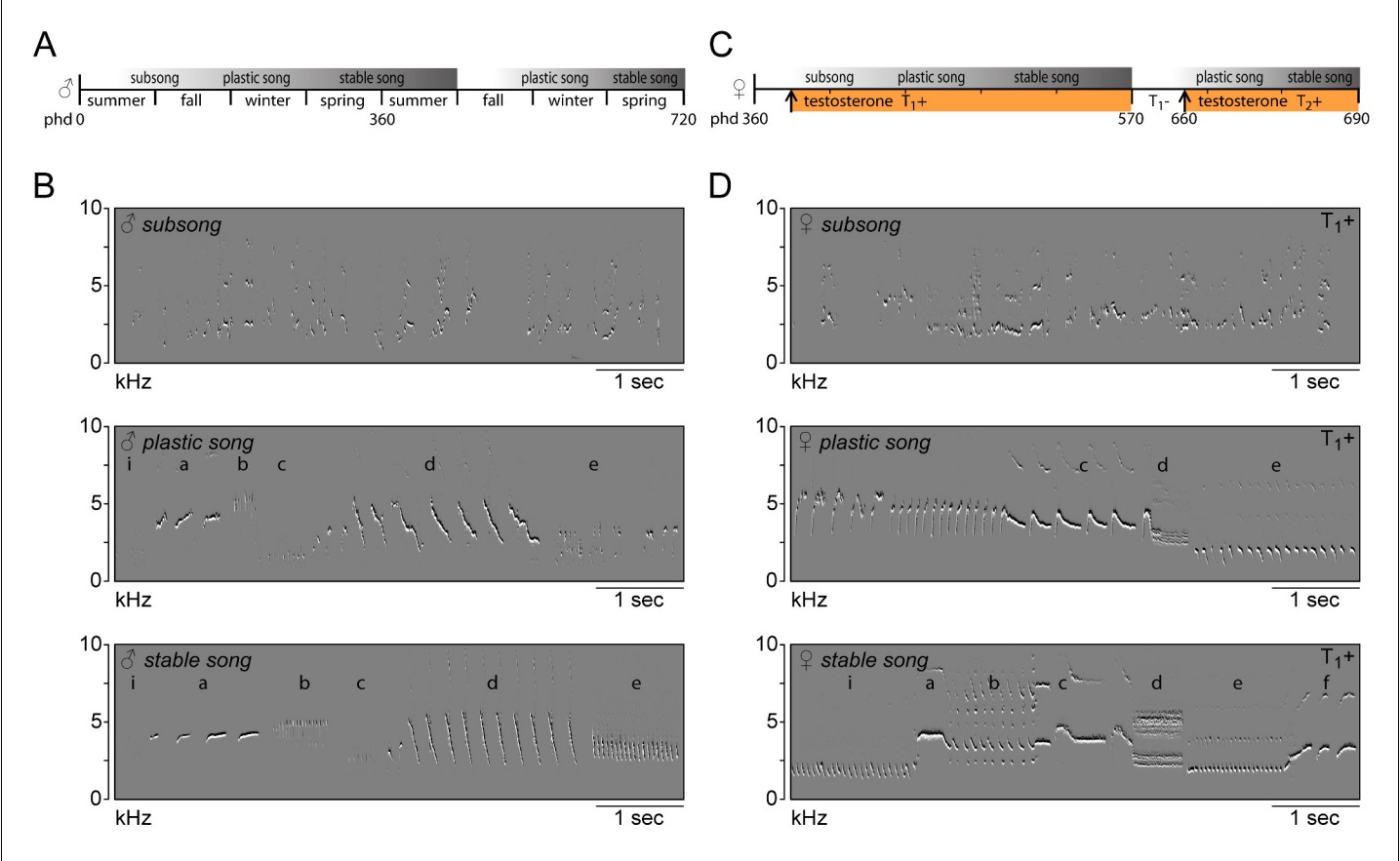

**Figure 1.** Song in juvenile male and adult female canaries progresses through the same stages of development. (A) Schematic of natural song development in juvenile males. (B) Example spectral derivative spectrograms illustrating the different song developmental stages of a male canary. Subsong was recorded at 45 days of age, plastic song at 120 days of age and stable song at 1 year old. (C) Schematic of song development in adult female canaries during a first testosterone treatment ($T_1+$), after removal of testosterone ($T_1-$), and during a second testosterone treatment ($T_2+$). (D) Example spectral derivative spectrograms from an adult female canary illustrating subsong after 3 days of testosterone treatment, plastic song after 30 days of treatment, and stable song after 200 days of treatment. Lowercase letters indicate different phrases of repeated syllables.

DOI: https://doi.org/10.7554/eLife.43194.003

$4.9 \pm 0.7$ Hz). A similar distribution of syllable rates redeveloped during a $2^{nd}$ testosterone treatment. The observed range of syllable rates was similar to what has previously been reported for wild-living male canaries (*Leitner et al., 2001*).

To determine how fast this syllable rate pattern emerged during subsequent testosterone treatments we took crystallized song from the last week of the first testosterone treatment as a reference pattern, and calculated the correlation coefficient (CC) between this reference and each day of song development and re-development (*Figure 2B & C*). While syllable repetition rates gradually developed and stabilized over the course of $160 \pm 8$ days during the first testosterone treatment, stable rates were obtained more than seven times faster, within $22 \pm 2$ days, during the second treatment (*Figure 2D*; paired t-test: p<0.001). In addition the maximum daily increase in similarity ($d_{max}$) was also significantly higher during song re-development than during initial song development (*Figure 2E*; $d_{max}$: $0.021 \pm 0.005$ for $T_1+$, and $0.078 \pm 0.012$ for $T_2+$; paired t-test: p<0.05), indicating a much accelerated development of song performance in birds with previous singing experience.

Since syllable rate is a compound feature that is determined by both syllable duration and the interval between subsequent syllables, we analyzed syllable durations and pause durations separately. Syllable durations developed slowly during the initial song acquisition ($T_1+$), reaching stable values after $153 \pm 19$ days, while pause durations stabilized in $82 \pm 13$ days (*Figure 2—figure supplement 1*, *Figure 2—figure supplement 2*). Both temporal features re-emerged more quickly

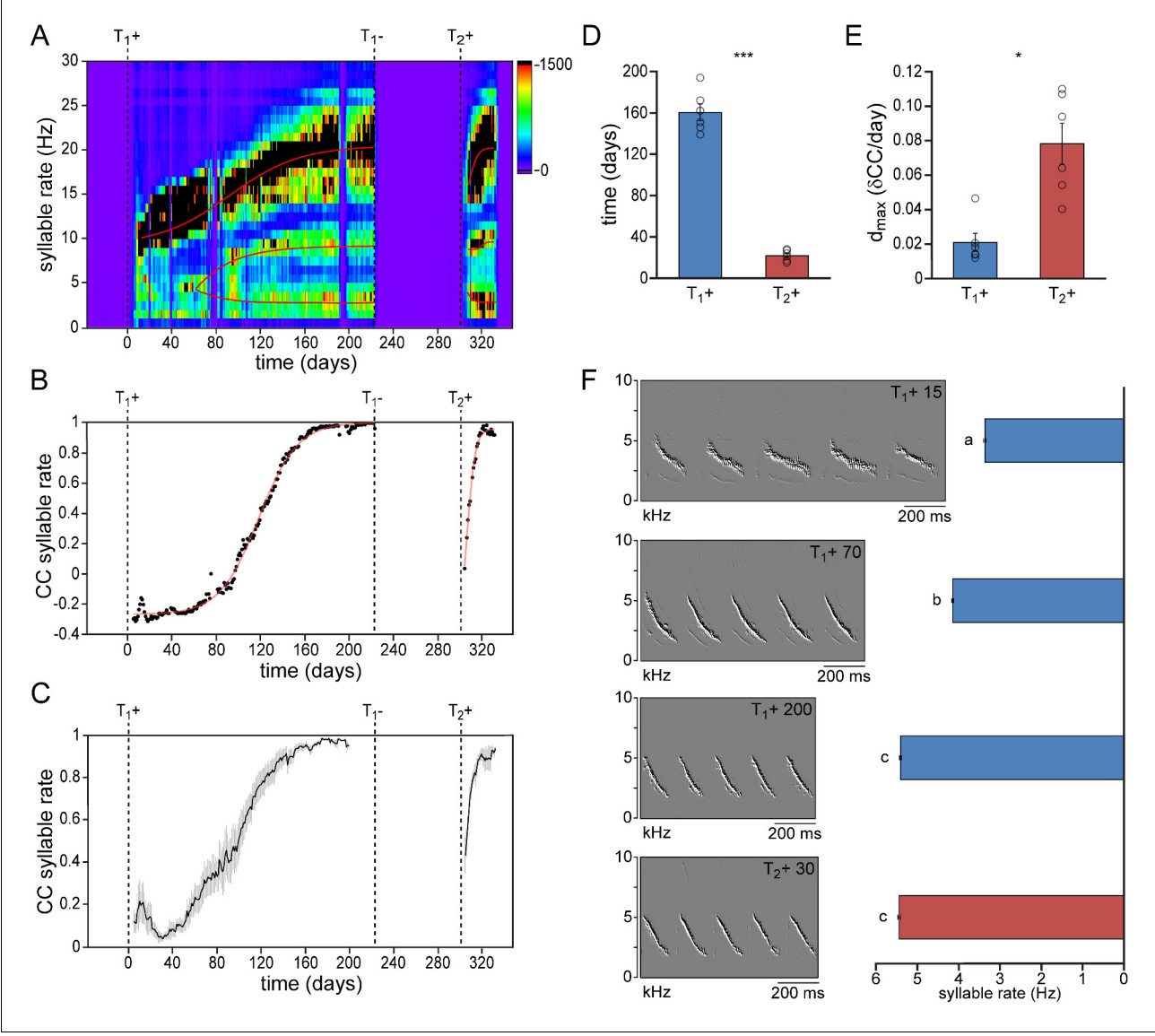

**Figure 2.** Syllable repetition rates develop faster in birds with prior singing experience. (A) Syllable rate histogram from a female canary during two subsequent testosterone treatments. Color scales indicate the number of daily syllables produced for each discrete syllable rate. Curves were fitted through the most occurring syllable rates and are shown in red. (B) The correlation coefficient (CC) between the stabilized distribution of syllable rates at the end of the 1st testosterone treatment and each other day during the development and re-development of song in the same bird as shown in A. (C) The mean syllable rate correlation plot for all animals. Grey bars indicate the SEM. (D) Group statistics of all studied birds demonstrating that stable syllable rates were achieved more quickly during a 2nd testosterone treatment (red bars) than during the 1st treatment (blue bars). (E) The peak day-to-day increase in the syllable rate CC ($d_{max}$) was higher during a 2nd testosterone treatment than during the 1st treatment. (F) Example spectral derivative spectrograms from one bird and corresponding bar graphs illustrating syllable rates at 15, 70, and 200 days after a 1st testosterone treatment and 30 days after a 2nd treatment. Columns in D-E represent the mean ± SEM and open circles indicate individual data points (*$p<0.05$, ***$p\leq0.001$, paired t-test, n = 6 animals). Columns in F represent the mean ± SEM (a,b,c: $p\leq0.001$, ANOVA; n = 100 syllables). Source data for temporal song features are available in the *Figure 2—source data 1*.

DOI: https://doi.org/10.7554/eLife.43194.004

The following source data and figure supplements are available for figure 2:

**Source data 1.** Source file for quantitative comparisons of temporal song features.
DOI: https://doi.org/10.7554/eLife.43194.007
**Figure supplement 1.** Differential re-development of temporal song features.
DOI: https://doi.org/10.7554/eLife.43194.005
**Figure supplement 2.** Development and re-development of temporal song features.
DOI: https://doi.org/10.7554/eLife.43194.006

during song re-acquisition (T$_2$+) than during the first song developmental phase (duration: 4.8 ± 1.3 days; paired t-test: p<0.001; pause: 10.7 ± 2.2 days; paired t-test: p<0.01; *Figure 2—figure supplement 1*).

To determine the bird's ability to recover spectral song patterns we analyzed the recurring patterns of frequency modulation (FM), amplitude modulation (AM), bandwidth (BW), mean frequency (MF) and Wiener entropy (E) during song acquisition and re-acquisition. Similar to temporal song features, spectral features were more quickly established during song re-acquisition (T$_2$+) than during the initial song development (T$_1$+) (*Figure 3*, *Figure 3—figure supplement 1*, *Figure 3—figure supplement 2*). All spectral features developed gradually over more than 110 days before reaching stable values. After a period in which the birds did not produce any song, testosterone-induced song re-acquisition led to a rapid stabilization of spectral patterns within 16 days (*Figure 3*, *Figure 3—figure supplement 1*; paired t test: p<0.01). Thus both temporal and spectral song features were acquired significantly faster by experienced birds that had developed singing skills once before.

## The absence of song practice causes selective deterioration of song features

Whereas song re-acquisition in female canaries resulted in a rapid recurrence of stable temporal and spectral patterns, not all song features re-developed in the same way. Most notably, some song features demonstrated a clear deterioration in the absence of vocal practice, while other song features remained stable despite the absence of vocal practice (*Figure 3G & H*).

Of the studied temporal song parameters, syllable durations did not show any deterioration in the period that birds did not sing (T$_1$-) and thus also did not need to be re-acquired (*Figure 2—figure supplement 1A–C*; *Figure 2—figure supplement 2A*). Both the syllable rate and the pause duration between syllables did deteriorate however, and a short phase of re-development was required to recover the originally developed patterns (*Figure 2*, *Figure 2—figure supplement 1D–F*, *Figure 2—figure supplement 2B*). The accelerated re-acquisition of syllable repetition rates thus appears to be driven by the need to re-optimize the timing between syllables rather than syllable lengths.

Of the spectral syllable features that were studied, the FM and AM patterns deteriorated when birds stopped singing (T$_1$-), but were re-acquired during a short developmental phase after a 2$^{nd}$ testosterone treatment (T$_2$+) (*Figure 3A–C*, *Figure 3—figure supplement 1A–C*, *Figure 3—figure supplement 2A & B*). BW, MF, and E did not need to be re-acquired, but maintained strong similarities with the initial acoustic patterns despite the absence of song production in the intermittent period (T$_1$-) (*Figure 3D–F*, *Figure 3—figure supplement 1D–I*, *Figure 3—figure supplement 2C–E*). Thus song re-acquisition can be characterized by recalling a combination of sound features that immediately can be reproduced and sound features that require a short phase of redevelopment.

## Song similarity during subsequent testosterone treatments

The songs that reappeared during the second testosterone treatment were strikingly similar to those that developed after the first treatment (*Figure 4*). Visual inspection of the song spectrograms revealed no differences in syllable repertoire when comparing crystallized songs during the first and second testosterone treatment (*Figure 4A & B*; 6.2 ± 1.1 syllable types for both treatments). In addition both temporal and spectral song features demonstrated a strong similarity in their distribution patterns between subsequent testosterone treatments (T$_1$×T$_2$), with correlation coefficients above 0.82 for all studied song features (*Figure 4C*). These values were not significantly different from the correlation coefficients obtained by cross-correlating distribution patterns from consecutive days during the first treatment period (T$_1$×T$_1$) for any of the measured song features (*Figure 4C*), indicating that the song patterns that developed during the first and second testosterone treatment are similar in structure.

To determine the similarity of syllables produced during the 1$^{st}$ and 2$^{nd}$ testosterone treatments we calculated for each bird the Euclidean distance across all eight analyzed temporal and spectral features within and between syllable types (*Figure 4—figure supplement 1*). The Euclidean distance between two syllables provides a measure of similarity where a syllable is represented as a point with coordinates that correspond to the eight analyzed sound features. The distance between

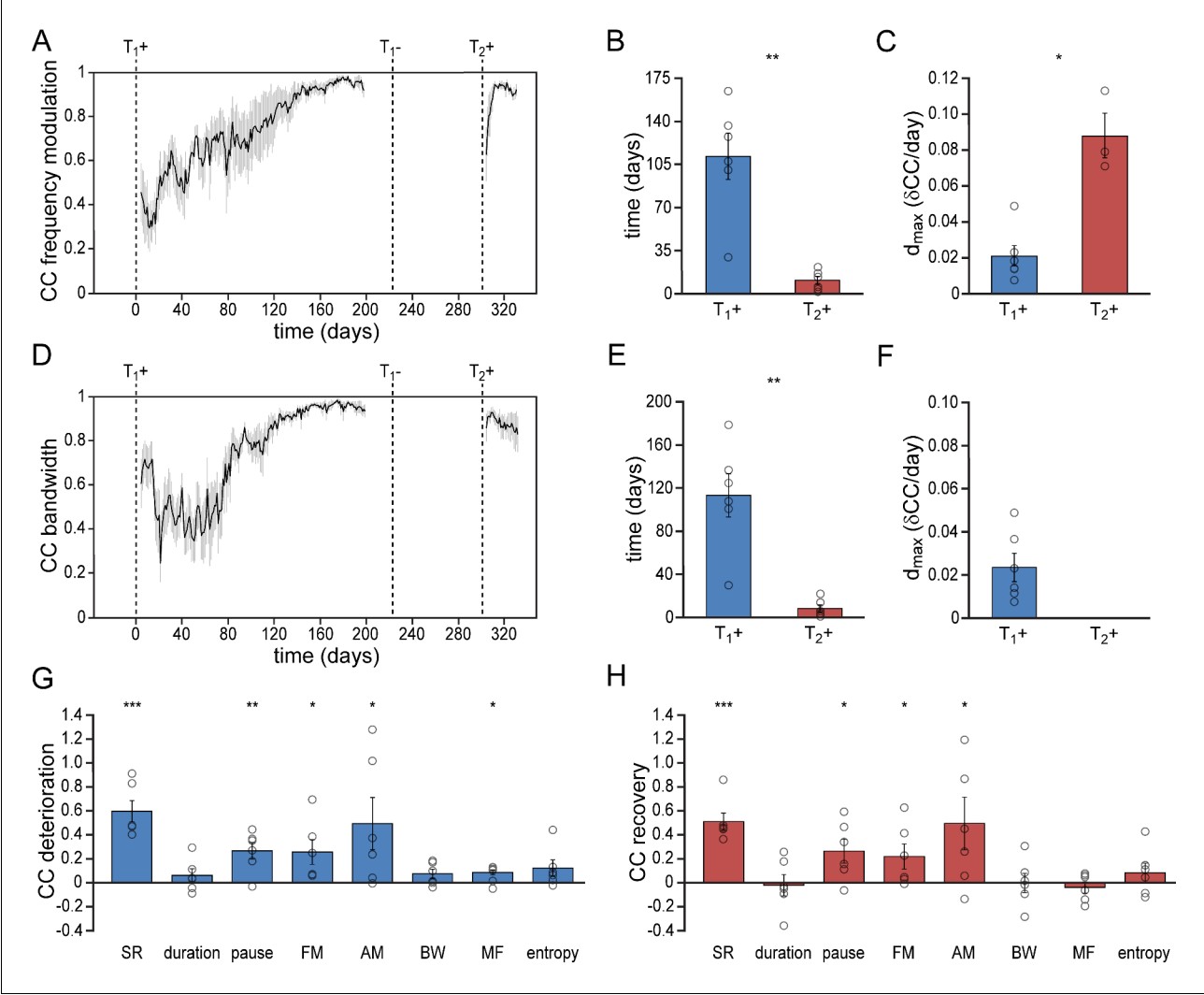

**Figure 3.** Differential re-development of song features. (A) Mean correlation plot for all animals illustrating the development of the frequency modulation (FM) distribution in the song during the development ($T_1+$) and re-development ($T_2+$) of song. The correlation plot illustrates a gradual FM development during $T_1+$, followed by a short phase of FM re-development during $T_2+$. (B) Group statistics demonstrating that the stabilization of the FM distribution in the song took less time during a 2nd testosterone treatment (red bars) than during the 1st treatment (blue bars). (C) The peak day-to-day increase in the FM CC ($d_{max}$) was significantly higher during a 2nd testosterone treatment than during the 1st treatment. (D) Mean correlation plot for the syllable bandwidth showing a gradual development during a 1st testosterone treatment ($T_1+$), followed by an immediate recovery of syllable bandwidth during a 2nd treatment ($T_2+$). (E) Group statistics demonstrating that stable syllable bandwidths were achieved more quickly during a 2nd testosterone treatment (red bars) than during the 1st treatment (blue bars). (F) The peak day-to-day increase in the syllable bandwidth CC ($d_{max}$) for $T_1+$. $D_{max}$ could not be calculated for $T_2+$, as we observed no developmental increase of this song feature during the 2nd testosterone treatment. (G) Deterioration in the distribution patterns of all analyzed song features during absence of song production ($T_1-$), and (H) subsequent recovery of song features during testosterone-induced re-development of song ($T_2+$). Grey bars in A and D indicate the SEM. Columns in B,C and E-H represent the mean ± SEM and open circles indicate individual data points (*p<0.05, **p<0.01, ***p≤0.001, paired t-test (B,C,E,F) and one-sample t-test (G,H), n = 6 animals). Source data for acoustic features are available in the *Figure 3—source data 1*.

DOI: https://doi.org/10.7554/eLife.43194.008

The following source data and figure supplements are available for figure 3:

**Source data 1.** Source file for quantitative comparisons of spectral song features.
DOI: https://doi.org/10.7554/eLife.43194.011
**Figure supplement 1.** Differential re-development of spectral song features.
DOI: https://doi.org/10.7554/eLife.43194.009
**Figure supplement 2.** Development and re-development of spectral song features.
DOI: https://doi.org/10.7554/eLife.43194.010

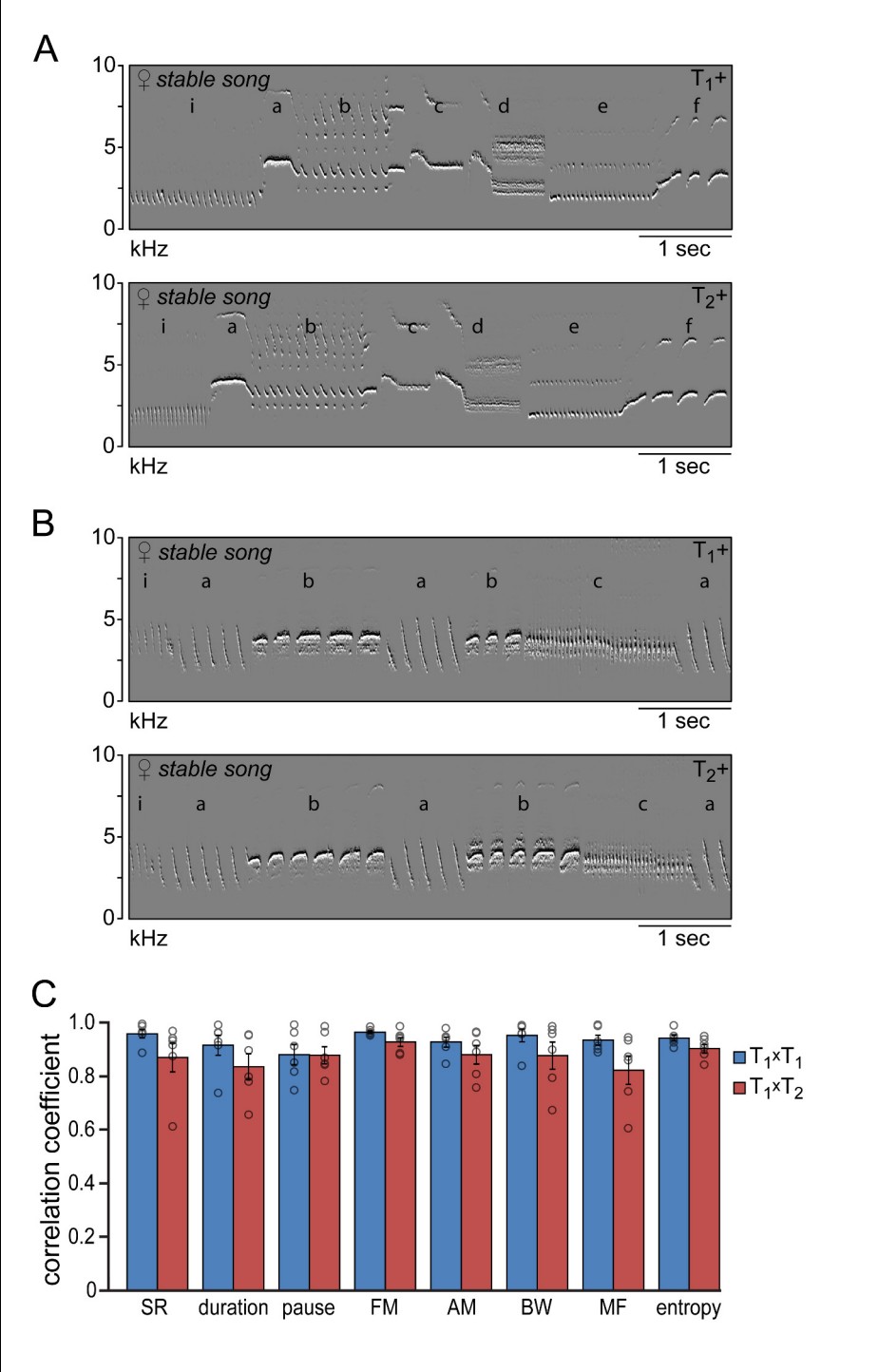

**Figure 4.** Similarity of time and frequency parameters after subsequent testosterone treatments. (**A,B**) Example spectral derivative spectrograms of stable song from two animals during the 1st ($T_1+$) and 2nd ($T_2+$) testosterone treatment illustrating a strong similarity in song structure. (**C**) Similarity analyses between stable song patterns from the 1st and 2nd testosterone treatment periods ($T_1xT_2$, red bars) demonstrated a high level of correlation of more than 80% for all analyzed song features. Correlation coefficients between songs from $T_1+$ and $T_2+$ were not significantly different from the CCs obtained when cross-correlating song patterns within the $T_1+$ period ($T_1xT_1$, blue bars). Columns in C represent the mean ± SEM and open circles indicate individual data points (NS, paired t-test, n = 6 animals). Source data for similarity calculations are available in the *Figure 4—source data 1*.
DOI: https://doi.org/10.7554/eLife.43194.012

*Figure 4 continued on next page*

*Figure 4 continued*

The following source data and figure supplements are available for figure 4:

**Source data 1.** Source file for song similarity calculations between subsequent testosterone treatments.
DOI: https://doi.org/10.7554/eLife.43194.015
**Figure supplement 1.** Syllable similarity during subsequent testosterone treatments.
DOI: https://doi.org/10.7554/eLife.43194.013
**Figure supplement 2.** Similarity of song syntax during subsequent testosterone treatments.
DOI: https://doi.org/10.7554/eLife.43194.014

syllables with similar sound features and thus similar coordinates is low, while the distance between syllables with differing sound features is high. The mean Euclidean distances between syllables of the same type across the 1$^{st}$ and 2$^{nd}$ testosterone treatment ($T_1{}^xT_2$) were not different from the Euclidean distances between syllables from subsequent days during the 1$^{st}$ testosterone treatment ($T_1{}^xT_1$: d = 0.73 ± 0.04, $T_1{}^xT_2$: d = 0.76 ± 0.06; Tukey's HSD: p=0.99). The mean Euclidean distances between syllables of different types across the two treatment periods were significantly larger than those between syllables of the same type ($T_1{}^xT_2{}^{ext}$: d = 2.12 ± 0.23; Tukey's HSD: p<0.001). These results indicate that for a given syllable the sound features that redeveloped after the 2$^{nd}$ testosterone treatment fell within the range of variation observed during the 1$^{st}$ testosterone treatment for that same syllable type, but were clearly distinguishable from the sound features of other syllable types.

To investigate if the sequential structure of the song was different between the two testosterone treatments we analyzed the probability that specific phrase transitions occurred in the songs (*Figure 4—figure supplement 2*). Because the syllable repertoire is relatively small in testosterone treated female canaries, the number of possible phrase transitions is also small. Nevertheless, on average only 8.7% of all possible phrase transitions were used in the songs. Thus whereas the song sequences are not fixed as in for example zebra finches, the song sequences are not random either, following a limited subset of possible transitions similar to what has been observed in male canaries (*Leitner et al., 2001*; *Markowitz et al., 2013*). The percentage of phrase transitions that made up the songs during the 1$^{st}$ and 2$^{nd}$ testosterone treatments were not different ($T_1+$: 9.2 ± 4.5%; $T_2+$: 8.3 ± 4.2%; paired t-test: p=0.59), and the probability distribution of these transitions were visually similar between treatments (*Figure 4—figure supplement 2A & B*). The linearity, consistency, and entropy of song phrase transitions did not differ significantly between the songs from the 1$^{st}$ and 2$^{nd}$ treatment periods (*Figure 4—figure supplement 2C*).

Together these data strongly suggest that testosterone-induced song re-acquisition in canaries is driven towards the previously acquired song pattern.

## Singing-related pruning of neuronal dendritic spines

The finding that the re-acquisition of song performance progresses considerably faster than the initial acquisition and leads to the production of highly similar song patterns suggests that singing skills are retained during intermittent silent periods. Motor learning and memory are generally considered to rely on the maturation and consolidation of the synaptic connections within the neural circuitry that drives the behavior in question (*Changeux and Danchin, 1976*; *Yu and Zuo, 2011*; *Hoshiba et al., 2017*). To investigate changes in synaptic connectivity during vocal motor development we quantified the number of neuronal dendritic spines in the forebrain motor nucleus HVC and the robust motor nucleus of the arcopallium (RA) before, during, and after the acquisition of vocal motor skills (*Figure 5*).

Excitatory projection neurons in the HVC of testosterone-treated, singing female canaries displayed a more than 30% reduction in dendritic spine density compared to untreated, non-singing control birds (C: 0.95 μm$^{-1}$, $T_1+$: 0.64 μm$^{-1}$; Dunnett's test: p<0.01), indicating a significant synaptic pruning during the first acquisition of singing skills (*Figure 5B*). Compared to non-singing control birds, spine densities remained significantly lower in birds that stopped singing for 2.5 months after testosterone removal (C: 0.95 μm$^{-1}$, $T_1-$: 0.69 μm$^{-1}$; Dunnett's test: p<0.01). Thus, the synaptic pruning in HVC associated with testosterone-induced song acquisition is not reversed when birds stop singing, but instead the pruned state is maintained in periods in which the birds do not sing

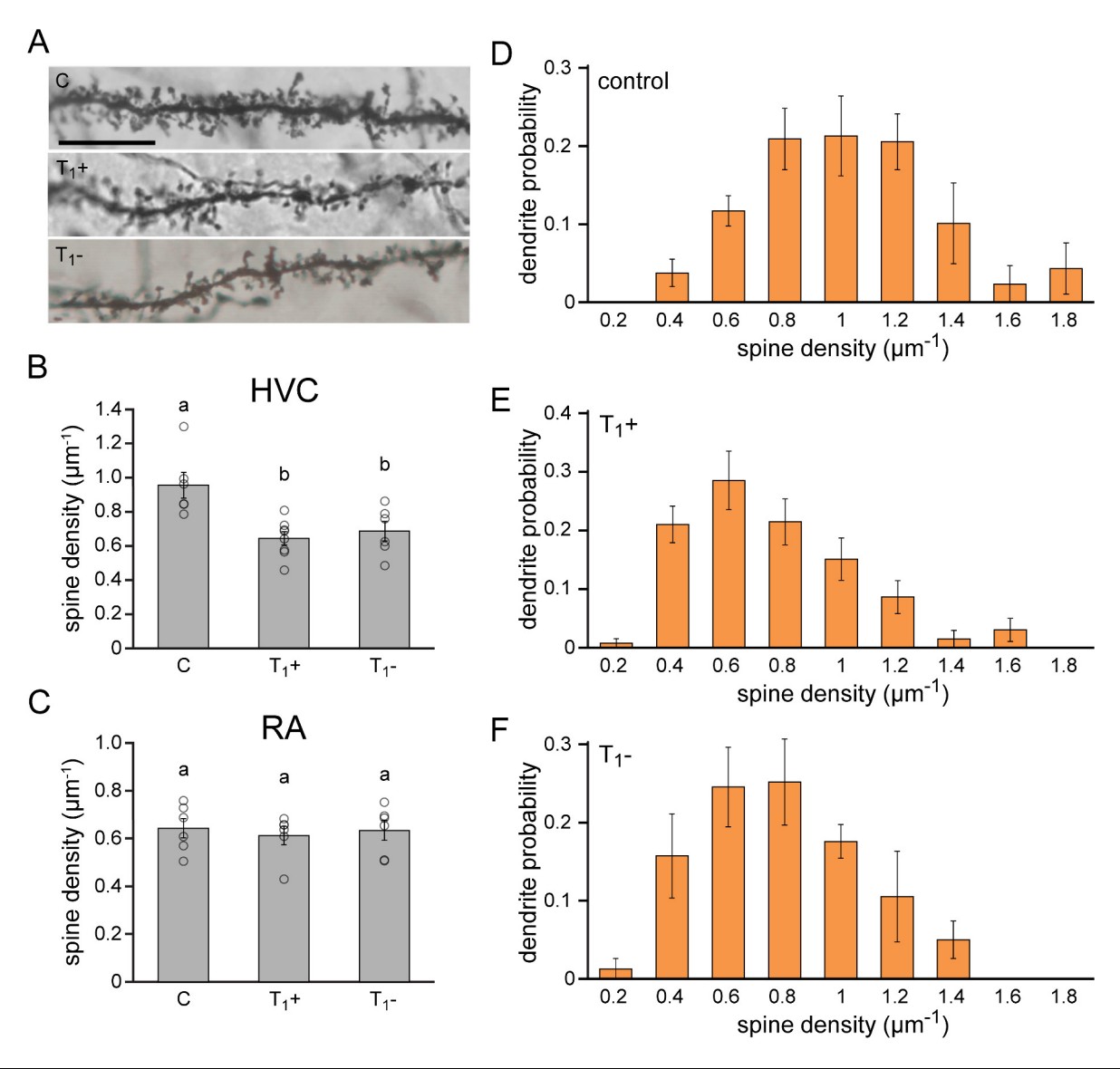

**Figure 5.** Lasting synaptic pruning in the forebrain motor nucleus HVC. (A) Photomicrographs of dendrite segments from non-singing control animals (C), singing female canaries sacrificed after five months of testosterone treatment (T$_1$+), and non-singing individuals sacrificed 2.5 months after testosterone withdrawal (T$_1$-). (B) Compared to naive control birds (C), spine densities were significantly reduced in testosterone-treated, singing birds (T$_1$+), and remained significantly reduced up to 2.5 months after birds stopped singing by withdrawing testosterone (T$_1$-). (C) No significant differences in spine densities were observed in RA between the experimental periods. (D–F) The probability distribution of dendrites with different spine densities in HVC demonstrated a shift towards more dendrites with fewer spines in testosterone treated (T$_1$+), singing birds and testosterone removed (T$_1$-), non-singing birds compared to non-singing control birds. Columns in B-F represent the mean ± SEM and open circles indicate individual data points (a,b: p<0.01, ANOVA, n = 6 animals). Scale bar = 100 µm. Source data for spine quantifications are available in the *Figure 5—source data 1*.

DOI: https://doi.org/10.7554/eLife.43194.016

The following source data is available for figure 5:

**Source data 1.** Source file for spine quantification data.

DOI: https://doi.org/10.7554/eLife.43194.017

(*Figure 5B*). In contrast, we did not observe any changes in dendritic spine densities in RA across the different experimental periods (*Figure 5C*; ANOVA: p=0.857).

Strong differences were observed in the density of dendritic spines between different spinous neurites in HVC (range: 0.15 to 2.14 spines/µm). To investigate which neurite types were associated

with the observed spine pruning we calculated the probability distribution of neuronal dendrites in HVC based on their spine density (*Figure 5D–F*). We observed a marked shift in the distribution of dendrites from more densely-spined dendrites in control birds towards less densely-spined dendrites in the testosterone treatment ($T_1+$) and removal groups ($T_1$-), reflecting a global decrease in spine density. The strongest reduction was detected in the range of dendrites with more than 0.8 spines per μm, which halved in number in both testosterone-treated, singing birds ($T_1+$), and testosterone-removed, non-singing birds ($T_1$-) compared to non-singing controls (C: $0.64 \pm 0.06$, $T_1+$: $0.28 \pm 0.05$, $T_1$-: $0.33 \pm 0.06$; ANOVA: $p<0.01$). The total density of both spinous and aspinous neurites in HVC did not significantly differ between the treatment groups (ANOVA: $p>0.36$), suggesting that changes in dendrite densities were not caused by specific cell death.

These data suggest that the observed vocal motor acquisition is accompanied by a lasting synaptic pruning specifically in nucleus HVC, providing a possible anatomical site for motor memory that could facilitate the observed rapid re-acquisition of previously established song patterns.

## Discussion

The ability to learn new skills enables one to adapt to changes in the surrounding environment. Whereas learning new skills requires time and effort, the survival of many species relies on their capacity to utilize specialized skills in a timely fashion, capitalizing on the short-term availability of foods, mates, and other environmental factors (*Prendergast et al., 2002*). Such a capacity is biologically relevant for numerous skills in a multitude of species, ranging from periodic foraging skills in dolphins (*Patterson et al., 2016*), bats (*Clarin et al., 2013*), and monkeys (*Boinski and Fragaszy, 1989*) to courtship displays in crabs (*Mowles et al., 2017*) and songbirds (*Ota et al., 2015*). As the use of these specialized skills is interrupted over longer periods, for example during long-distance migration or hibernation (*Slagsvold, 1976*; *Prendergast et al., 2002*), such skills need to be re-acquired quickly to maximize survival and reproductive success. In this study we establish the repeated, hormone-driven acquisition of vocal motor performance in female canaries as a model system for studying motor savings. Using this model, we show that while the initial development of specialized motor skills can be a lengthy process, motor performance can quickly be re-acquired at a later time, even after extended periods of non-use (*Figure 2 and 3*). The accelerated song re-acquisition observed in this study indicates that acquiring a motor skill for the first time follows a different developmental trajectory than re-acquiring the same skill at a later time. This is both evident from the accelerated consolidation of song performance during song re-development (*Figure 2 and 3*), and the observation that birds skip the subsong phase of song development when re-acquiring their songs. Our data further suggest that songbirds have the ability to retain previously developed singing skills through a process of selective pruning of the neural circuitry (*Figure 5*), enabling them to rapidly recover songs of adequate quality to attract a mate before breeding opportunities disappear.

Several song parameters were maintained across a period in which birds did not sing, while other song features demonstrated clear deterioration (*Figure 3G & H*). Birds were able to recover those deteriorated song features with renewed vocal practice. At the peripheral level, song performance depends on the accurate control of the respiratory system, the sound producing organ (the syrinx), and the upper vocal tract. Interestingly, those song features that required a phase of re-development to be optimized seem primarily associated with motor constraints in syrinx function (*Podos, 1997*; *Geberzahn and Aubin, 2014*). Online modulation of frequency and amplitude requires active contractile modulation of syringeal muscles (*Goller and Suthers, 1996*; *Elemans et al., 2008*). Syllable repetition and the inter-syllabic interval also depend on how rapidly syringeal abductor and adductor muscles can open and close the syringeal aperture. The upper vocal tract may act as a tunable acoustic filter of the sounds produced by the syrinx (*Nowicki, 1987*; *Beckers et al., 2003*; *Riede et al., 2006*) and play a more important role in establishing the bandwidth, entropy and mean frequency of song syllables. Thus song parameters linked to vocal tract modulation may be preserved after birds stopped singing, while song parameters linked to contraction speed of syringeal muscles appear to deteriorate.

Whereas male canaries are known to annually re-organize their songs, even when deafened (*Mori et al., 2018*), no such modifications were observed in our study. It is not clear to what extent annual changes in male canary song structure are due to relearning or are driven by a selection

process from a previously learned collection of syllable types during juvenile development (*Nottebohm et al., 1986*; *Marler, 1997*; *Gardner et al., 2005*; *Belzner et al., 2009*). Considering that the female canaries in our study were acoustically separated during the testosterone-induced development of song suggests that the emerging song patterns are either for a large part innate (*Mori et al., 2018*; *Gardner et al., 2005*), or are limited to a memorized collection of syllable types that they heard earlier in life (*Belzner et al., 2009*; *Marler and Peters, 1981*; *Marler and Peters, 1982b*; *Prather et al., 2010*; *Kiefer et al., 2014*). The initial song acquisition may thus rely on an innate or memorized auditory template, while the accelerated song re-acquisition may be driven by a motor memory trace of what the birds learned to produce during the first song acquisition phase. The syllable repertoires of our female canaries were much lower than what is common for males. Such a small vocabulary would limit the possibilities to re-arrange or replace syllables without compromising song complexity, and thus few modifications to female songs can be expected.

During juvenile brain development in humans, mammals, and birds, synaptic connections are initially overproduced, followed by protracted, activity-dependent synapse elimination during puberty (*Huttenlocher, 1990*; *Petanjek et al., 2011*; *Rakic et al., 1986*; *Zuo et al., 2005*; *Nixdorf-Bergweiler et al., 1995*). By selectively stabilizing functional synapses while pruning away inactive synapses, brain circuits are thought to reduce neuronal plasticity, consolidating newly learned information and reducing unnecessary redundancy (*Changeux and Danchin, 1976*). The lasting spine pruning observed in the forebrain song control area HVC (*Figure 5*) could reflect such a structural modification facilitating the long-term memorization of vocal patterns. Optical imaging studies have demonstrated a rapid accumulation and stabilization of neuronal dendritic spines in HVC during developmental song learning in zebra finches (*Roberts et al., 2010*). In the mouse cortex, morphological modifications to dendritic spines have been observed that lasted well beyond the learning experience that caused the spine modifications (*Hofer et al., 2006*; *Yang et al., 2009*; *Xu et al., 2009*; *Hofer et al., 2009*), strongly suggesting that long-term memory retention of previously acquired skills is facilitated by stably maintained synaptic connections. Despite such learning-related circuit modifications, the large majority of dendritic spines in the mammalian brain are stably maintained in adulthood (*Petanjek et al., 2011*; *Rakic et al., 1986*; *Zuo et al., 2005*; *Yang et al., 2009*; *Xu et al., 2009*; *Grutzendler et al., 2002*). In contrast we observed a strong spine reduction in fully adult birds (*Figure 5*), suggesting that the observed synaptic pruning in the adult canary HVC during song acquisition may reflect a developmentally delayed, activity-dependent consolidation of the neural motor circuitry.

A recent study in zebra finches has shown that singing prevention during juvenile development can also avert developmental spine pruning in the robust motor nucleus of the arcopallium (RA) (*Hayase et al., 2018*). In addition, auditory input during sensorimotor learning may play an important role in circuit consolidation, and auditory deprivation has been shown to delay the natural juvenile maturation of HVC in zebra finches (*Huang et al., 2018*). It is not known how long such developmental processes can be postponed, but we did not observe delayed synaptic pruning in RA upon song acquisition in adult female canaries (*Figure 5C*), suggesting that the time-window for large-scale spine pruning in RA may be more limited than in HVC. The developmentally delayed maturation of HVC (*Figure 5B*) is consistent with our observation that the delayed, testosterone-induced development of song performance in adult female canaries follows a juvenile male-like developmental trajectory (*Figure 1*). Thus, in the absence of singing behavior, parts of the vocal motor circuit may remain plastic in adult female canaries, only consolidating once neural activity within the circuitry increases to drive testosterone-induced song output.

Testosterone-induced song development in adult female canaries has previously been associated with changes in other neural attributes in HVC as well, including an increase in volume, neuron numbers (but not densities), increased vascularization, and changes in cellular structure (*Vellema et al., 2014*; *Hartog et al., 2009*; *Gahr and Garcia-Segura, 1996*; *Nottebohm, 1980*; *Rasika et al., 1994*; *Louissaint et al., 2002*). Many of the observed modifications to the song system follow an annually changing pattern in seasonally breeding male songbirds (*Nottebohm et al., 1986*; *Smith et al., 1997*; *Nottebohm, 1981*; *Kirn et al., 1994*), indicating that in males these modifications are not maintained outside the breeding season when birds no longer sing high performance song. In addition, withdrawing testosterone from male Gambel's white-crowned sparrows, another well-studied seasonal singer, resulted in a rapid regression in HVC size, neuron number, and neuron size (*Thompson et al., 2007*), indicating that these neural modifications are reversible and unlikely to

reflect long-term memorization of vocal patterns. The lasting synaptic pruning that we observed in HVC is to our knowledge the first report of a song-related modification to the adult songbird brain that persists after the behavior itself is no longer expressed.

HVC is at a central position between the motor circuitry that controls song output and a forebrain-basal ganglia circuit, analogous to the mammalian cortico-basal ganglia (CBG) circuit, which feeds back on the motor circuitry for song production (*Brainard and Doupe, 2013*; *Nottebohm et al., 1976*). Firing patterns in HVC neurons that project to RA are closely associated with millisecond precision of syllable timing (*Yu and Margoliash, 1996*; *Hahnloser et al., 2002*; *Amador et al., 2013*), suggesting that HVC is directly involved in maintaining time-related features of song performance. Burst patterns in RA neurons on the other hand are more closely associated with intrasyllabic structure (*Yu and Margoliash, 1996*; *Chi and Margoliash, 2001*). However, slowing down brain processes by local cooling of HVC has also been shown to change the intrasyllabic structure (*Long and Fee, 2008*; *Alonso et al., 2015*), indicating that both HVC and RA activity play a role in maintaining the fine temporal structure within syllables. The acquisition of song syllables in zebra finches is accompanied by increased timing accuracy of inhibitory neural firing within the HVC microcircuitry, suggesting that precisely-timed inhibition may play an important role in shaping song-related premotor sequences (*Kosche et al., 2015*; *Vallentin et al., 2016*). The lasting synaptic reorganization in HVC during song acquisition may be guided by such inhibitory activity and provide a neural scaffold for both intra and intersyllabic timing. RA-projecting HVC neurons are continuously replaced in the adult songbird brain (*Alvarez-Buylla et al., 1990*), providing a base for ongoing variability in syringeal muscle control. A permanently maintained HVC network may guide dynamically changing circuits in HVC and RA during subsequent periods of singing, quickly driving song features towards previously acquired patterns.

Neurons from HVC that project to the avian basal ganglia Area X are not replaced in adulthood, providing a permanent subpopulation of HVC neurons (*Gahr, 1990*; *Alvarez-Buylla et al., 1988*). Although we could not specifically distinguish between neuron types in our study, we observed the strongest reduction of 50% in the number of densely-spined dendrites in HVC after birds developed song. Previous studies have shown that those HVC neurons that project to Area X predominantly consist of neurons with densely-spined dendrites (*Kornfeld et al., 2017*; *Nixdorf et al., 1989*; *Kubota and Taniguchi, 1998*; *Benton et al., 1998*; *Mooney, 2000*; *Mooney and Prather, 2005*; *Fortune and Margoliash, 1995*). Thus our observed synaptic pruning may be related to a major reorganization within the HVC to Area X connection, a reorganization that could reflect the long-term memorization of vocal skills. CBG-like circuits have been strongly implicated in the learning and long-term retention of motor skills in humans (*Albouy et al., 2008*; *Debas et al., 2010*; *Lehéricy et al., 2006*; *Poldrack et al., 2005*; *Coynel et al., 2010*), mammals (*Barnes et al., 2005*; *Pasupathy and Miller, 2005*; *Sheth et al., 2011*; *Yin et al., 2009*), and birds (*Bottjer et al., 1984*; *Scharff and Nottebohm, 1991*; *Brainard and Doupe, 2000*; *Kao et al., 2005*; *Olveczky et al., 2005*; *Aronov et al., 2008*; *Sober and Brainard, 2009*; *Andalman and Fee, 2009*; *Ölveczky et al., 2011*; *Charlesworth et al., 2012*). Vocal exploration is driven by activity in the avian CBG circuit (*Kao et al., 2005*), which may guide synaptic modifications in plastic motor circuits (*Mehaffey and Doupe, 2015*), even in adulthood. Although selective elimination of X-projecting HVC neurons did not appear to alter the song structure of adult zebra finches (*Scharff et al., 2000*), altered expression of FoxP2 in Area X of adult male zebra finches has been associated with changes in vocal performance (*Murugan et al., 2013*; *Thompson et al., 2013*; *Teramitsu and White, 2006*). In addition neurotoxic lesions in Area X lead to changes in singing tempo and syllable sequencing in adult zebra finches (*Kubikova et al., 2014*), suggesting that the HVC to Area X connection continues to play a role in the long-term maintenance of vocal sequences.

The mechanisms behind learning-related, selective stabilization and elimination of dendritic spines remain largely unknown. Sex hormones may play an important role in the stabilization of such synaptic connections during the development of motor skills. Many of the brain areas that are involved in song learning and production express high numbers of androgen receptors (*Arnold et al., 1976*; *Balthazart et al., 1992*; *Metzdorf et al., 1999*; *Nastiuk and Clayton, 1995*; *Gahr and Metzdorf, 1997*), and X-projecting HVC neurons express estrogen receptors (*Gahr, 1990*) that can be activated after conversion of testosterone into estradiol. Furthermore, changes in neuronal dendritic growth and synaptic morphology in the songbird RA have been observed to accompany testosterone-induced song development in female canaries (*DeVoogd and Nottebohm, 1981*;

*Devoogd et al., 1985*; *Canady et al., 1988*). In addition, Frankl-Vilches et al. (*Frankl-Vilches et al., 2015*) recently observed that 85% of the seasonally and testosterone upregulated genes in the canary HVC are related to neuron differentiation, axon, dendrite and synapse organization. In the brain of birds and mammals, sex hormones are known to upregulate the expression of brain-derived neurotrophic factor (BDNF) (*Dittrich et al., 1999*; *Rasika et al., 1999*; *Sohrabji et al., 1995*), which in its turn contributes to the stabilization of synapses through the interaction with tyrosine receptor kinase B (TrkB) (*Koleske, 2013*). Furthermore, BDNF has been shown to be essential during testosterone-induced song development in female canaries (*Hartog et al., 2009*), providing a likely pathway by which sex hormones can consolidate motor skills by stabilizing synapses.

Testosterone levels and song consolidation are tightly linked, and it is difficult to determine if brain anatomical changes in our and other studies are caused by testosterone, singing activity or both. Interestingly, local intracerebral testosterone implants close to nucleus RA did not cause anatomical changes in this androgen receptor-rich brain area (*Brenowitz and Lent, 2002*), suggesting that modifications in RA are driven by singing-related, activity-dependent input from HVC. Whether or not testosterone, vocal practice, or both play a role in the observed synaptic pruning in HVC, the vocal circuitry demonstrated synaptic changes that persisted through several months in which testosterone levels were low and birds did not sing, potentially enabling the birds to quickly re-acquire their previously developed song patterns.

Sex hormones can be regulated by both environmental and social factors (*Goymann et al., 2007*; *Oliveira, 2004*; *Wingfield et al., 1990*), thereby providing a timed cue for the consolidation of behavioral output and the underlying brain circuitry. Such a cue could facilitate the timely formation and recall of long-term memories of relevant sensory and motor information. Our study suggests that lasting, hormone-driven synaptic pruning of related brain circuitries could form a basis for such long-term memories, enabling a quick recovery of previously acquired skills. The ability to respond quickly to environmental and social cues through such primed hormone-sensitive control mechanisms would constitute an important adaptation to ensure competitive proficiency and maximize reproductive success under environmental restrictions. The results obtained from this study may have implications not only across different animal groups, but may generally apply to different motor, sensory, and motivational systems that rely on a rapid recall of previously learned information.

## Materials and methods

### Subjects

For this study adult domesticated canaries (*Serinus canaria*) were either purchased from a local breeder in Antwerp or taken from the breeding colony of the Max Planck Institute for Ornithology, Seewiesen. All animals were raised and kept under local natural daylight conditions, until the day length reached 11 hr in early March. Birds were subsequently housed individually in sound attenuating chambers for song recordings. Experimental procedures were conducted according to the guidelines of the Federation of European Animal Science Associations (FELASA) and approved by the Ethical Committee on animal experiments of the University of Antwerp.

### Hormone treatment

To stimulate song production in naive female canaries, birds were subcutaneously implanted ($T_1$+) with 8 mm silastic tubes (Dow Corning, Midland, MI; ID: 1.47 mm) filled with crystalline testosterone (Sigma-Aldrich Co., St. Louis, MO). During the hormone treatments a light cycle of 11 hr light, and 13 hr darkness was maintained to exclude photoperiodic effects on song production. This light cycle was chosen because under natural conditions it corresponds to a phase of testosterone up-regulation and physiological sensitivity to hormone fluctuations in canaries (*Nottebohm et al., 1987*). After consolidation of song we removed the hormone implants ($T_1$-) to end song production, and changed the daylight conditions to 8 hr light and 16 hr dark, inducing a molting phase of approximately 1.5 months. Once the molt was finished, the day length was gradually returned to 11 hr over the following month. Subsequently, the song-experienced birds received a second testosterone treatment ($T_2$+), and were sacrificed 5 weeks after implantation.

## Hormone analysis

Blood samples were collected from the birds' right wing veins using heparinized haematocrit capillaries (Brand, Wertheim, Germany). Directly after blood collection, samples were centrifuged at 3000 RPM for 10 min to separate cells from plasma, and stored at −80°C for later analysis.

Testosterone concentrations in the blood plasma were determined by radioimmunoassay (RIA) after extraction and partial purification on diatomaceous earth (glycol) columns, following the procedures described previously (*Goymann et al., 2001*; *Goymann et al., 2008*).

Briefly, plasma samples were extracted with dichloromethane (DCM) after overnight equilibration of the plasma with 1500 dpm of $^3$H-labeled testosterone (Perkin-Elmer, Rodgau, Germany). After separation of the organic phase, the extracts were re-suspended in $2 \times 250$ µl 2% ethylacetate in isooctane and fractioned using columns of diatomaceous earth:propylene glycol:ethylene glycol (6:1.5:1.5, w:v:v). Collected testosterone fractions were dried, re-suspended in phosphate-buffered saline with 1% gelatin (PBSG), and left to equilibrate overnight at 4°C. Extraction of plasma testosterone with dichloromethane resulted in $86 \pm 11\%$ (mean ± sd) recovery. Hormone samples were incubated with antisera against testosterone (Esoterix Endocrinology, Calabasas Hills, CA), and after 30 min $^3$H-labeled steroids (13500 dpm) were added and left to incubate for 20 hr at 4°C. Free steroids were separated from the bound fractions by absorption on dextran-coated charcoal, centrifuged, and decanted into scintillation vials to be counted.

Testosterone concentrations in the plasma samples were measured in one assay. The lower detection limit of the RIA was 0.33 pg/tube, and all measured plasma levels were above the lower detection limit. The intra-assay variation of a chicken plasma pool as control sample at the beginning and end of the assay was 6.7%. Pooled plasma levels of testosterone for all birds during the different consecutive hormone treatments are shown in *Table 1*. Mean detected hormone levels were within range of physiological plasma levels in nest-building male canaries (range: 360–7970 pg/ml; n = 6) and previously reported values for other small passerines (*Wingfield and Hahn, 1994*; *Kempenaers et al., 2008*; *Apfelbeck and Goymann, 2011*).

## Song recording

To monitor the entire song ontogeny during consecutive testosterone treatments randomly selected, naive female canaries (n = 6) were kept individually in sound attenuating chambers during testosterone treatment ($T_1+$), testosterone removal ($T_1-$) and testosterone re-treatment ($T_2+$), while song output was continuously recorded. This approach resulted in a song database of $5.6 \pm 0.6$ million song syllables per bird, recorded over the course of approximately 1 year.

The vocal activity of each bird was recorded using Sound Analysis Pro 2.0 (*Tchernichovski et al., 2000*). Omnidirectional condenser microphones (TC-20; Earthworks, Milford, NH) connected to a multi-channel microphone preamplifier (UA-1000; Roland, Los Angeles, CA) were used to acquire and digitize all sounds produced within the sound attenuating boxes with a sampling frequency of 44.1 kHz. The incoming signal was filtered online using an amplitude and Wiener entropy threshold to exclude background noises, and saved in 60 s waveform audio files (16-bit PCM format).

## Song analysis

Vocalizations were segmented into individual syllables with the fully automated Feature Batch module in Sound Analysis Pro by applying an amplitude threshold to the sound wave. Because the

**Table 1.** Plasma levels of testosterone (T) during different hormone treatments.

| Treatment | T levels (pg/ml) | Significance* |
|---|---|---|
| control | 238 ± 65 | |
| $T_1+$ | 6486 ± 488 | p<0.001 |
| $T_1-$ | 138 ± 30 | NS |
| $T_2+$ | 7279 ± 1249 | p<0.01 |

\* P-values of comparisons between different hormone treatments against control values (ANOVA with Dunnett's t correction for multiple comparisons against one control).

DOI: https://doi.org/10.7554/eLife.43194.018

distance from perch to microphone ranged between 10 and 20 cm only, little variation in recorded amplitude levels can be expected. The amplitude threshold was selected once manually to assure reliable segmentation, and was kept constant during the analyses of all sound phrases within birds.

To filter out non-song vocalizations, syllables were only included when produced within a song bout of at least 750 ms. A song bout was defined as a sequence of sounds traversing the amplitude threshold with an interval of no more than 100 ms. We used Sound Analysis Pro to measure duration ($d$), pause duration ($i$), syllable rate (SR), frequency modulation (FM), amplitude modulation (AM), syllable bandwidth (BW), mean frequency (MF) and Wiener entropy (E) for each syllable and stored these syllable features in MySQL 5.1 tables (Oracle, Redwood Shores, CA). Bandwidth was calculated for each syllable by subtracting the minimum peak frequency from the maximum peak frequency. The syllable rate was defined for each syllable as: $SR_a = (d_a + i_a)^{-1}$, where $d_a$ is the duration of syllable 'a', and $i_a$ is the interval between the end of syllable 'a' and the beginning of the consecutive syllable within the same song bout, irrespective of syllable type. For a detailed computational description of how Sound Analysis Pro calculates the different sound features we refer to the accompanying publication (*Tchernichovski et al., 2000*) and the online manual (http://soundanalysispro. com).

To investigate the dynamic change of song features over the course of the experiment, daily histograms were obtained from the entire dataset by rounding individual values to the nearest integer, and plotting the number of times those integers occurred each day. To determine the SRs that were most commonly used at the onset and crystallization of song development a non-linear exponential curve (1) was fitted through the peak values of each SR histogram.

$$F(x) = a + \frac{b}{1 + e^{\left(\frac{c-x}{d}\right)}} \tag{1}$$

Further developmental correlation plots for all temporal and spectral song features were produced by calculating the Pearson product-moment correlation coefficient (CC) between a 7 day average of the histogram pattern at the time of song stabilization and all other recorded days. This calculation resulted in a value between −1 and 1 for each recorded day, and was used as a measure of similarity where one describes an absolute copy of the stable song pattern, and −1 describes the absolute inverse of the stable song pattern. A non-linear exponential curve (1) was fitted through the CC values and song parameters were considered stable as soon as the daily increase in similarity dropped below 0.001. For each song feature we used the fit curve to determine the duration between testosterone implantation and feature stabilization, and to determine the maximum daily increase in similarity ($d_{max}$). $D_{max}$ could only be calculated for birds and song features that demonstrated a developmental increase in similarity, and was only used for statistical purposes if three or more birds demonstrated such an increase. Mean CC plots combining all analyzed birds were calculated by averaging the raw (non-fitted) CC values that were calculated for each bird and for each day. We aligned song development in the different birds based on the first day where we detected song.

Spectrograms of the original sound files were visually inspected on syllable-like sound structures to determine the syllable repertoire of each bird and the first occurrence of song after testosterone treatments. To assess syllable similarity during consecutive testosterone treatments we calculated the Euclidean distance between each of 100 randomly selected syllables for each syllable type from stable songs during both the $T_1+$ and $T_2+$ periods. The Euclidean distance between two syllables 'a' and 'b' with coordinates $c = SR, d, i, FM, AM, BW, MF, E$ was defined as:

$$d(a,b) = \sqrt{\sum_{c=1}^{n}(b_1 - a_1)^2}$$

To give all analyzed temporal and spectral features equal weight in the analysis, each feature was normalized by dividing its value by its global median value from all syllables produced by all birds recorded in this study.

First, for each syllable type the mean Euclidean distance between 100 syllables of the same type during $T_1+$ ($T_1{}^xT_1$) was calculated to determine how much baseline variation existed between syllables of the same type. Lower values indicate a higher similarity between syllables. Secondly, we calculated the mean Euclidean distance between 100 syllables of the same type from the $T_1+$ and $T_2+$

$(T_1{}^xT_2)$ periods to determine if syllables produced during the 2nd testosterone treatment were similar to those produced during the 1st testosterone treatment. Finally, we calculated the mean Euclidean distance between 100 syllables of the same type from the $T_1$ +period with each of 100 syllables from all other types of syllables that a bird produced $(T_1{}^xT_2{}^{ext})$ to provide a measure of dissimilarity between syllables of different types. All observed syllable types were included in this analysis with the exception of single transitional syllables connecting two song phrases.

To assess similarity of song syntax between subsequent testosterone treatments we determined the probability of song phrase transitions during the $T_1$+ and $T_2$+ periods for each bird. On average 794.3 phrase transitions per bird were used for the analysis. From the resulting transition probability distributions we calculated the sequence linearity, consistency and entropy as previously described (*Scharff and Nottebohm, 1991*; *Daou et al., 2012*). In short, sequence linearity is expressed as the number of syllable types divided by the number of transition types and addresses the way phrases are ordered in a song. Sequence consistency is expressed as the sum of common phrase transitions divided by the sum of total transitions and addresses how often the same sequences are produced. Common phrase transitions were defined as transitions that make up at least 5% of all phrase transitions. The entropy (S) of the transition probability distribution with transition type $T_i$ is a measure of the spread of that distribution and was calculated as:

$$S = -\sum_{i=1}^{n} T_i \log_2(T_i)$$

## Specimen preparation

To estimate spine densities and dendrite densities in motor nucleus HVC and RA, brains were collected from a control group of adult female canaries that did not receive any testosterone treatment (C: n = 6), a group of birds that was sacrificed five months after testosterone treatment ($T_1$+: n = 8), and a group of birds that was sacrificed at least 2.5 months after testosterone withdrawal ($T_1$-: n = 6). All experimental birds were raised and kept in group aviaries together with other male and female canaries throughout the experiment to assure similar social and auditory conditions. Birds were transferred to sound attenuating chambers for two weeks after testosterone treatment to assure that birds were singing, and prior to sacrifice. Individuals were randomly allocated to the different treatment groups. All birds were maintained on a light cycle of 11 hr light, and 13 hr darkness and were between 2.5 and 3 years old at the time of sacrifice. Brains were processed for Golgi-Cox staining using the FD Rapid GolgiStain kit (FD NeuroTechnologies, Columbia, MD).

After overnight fixation in a 4% formaldehyde solution in phosphate-buffered saline (PBS; 10 mM; pH 7.4), brains were stored in PBS with 0.05% sodium azide at 4°C until further processing.

Brain hemispheres were separated with a razor blade and left hemispheres were immersed for two weeks in an impregnation solution consisting of equal amounts of FD solutions A and B in total darkness at room temperature. Following three days of incubation in FD solution C, brains were cut into 30 µm sagittal sections using a sliding microtome (Leica Microsystems GmbH, Wetzlar, Germany), which were stored in PBS with 0.05% sodium azide at 4°C.

Sections with a regular interval of 180 µm were mounted on SuperFrost glass slides (Menzel GmbH, Brauschweig, Germany), developed in a solution of two parts distilled water, one part FD solutions D, and one part FD solution E for ten minutes, and rinsed in distilled water before embedding in CC/Mount tissue mounting medium (Sigma-Aldrich). Slides were incubated at 70°C until the mounting medium had hardened, cleared in xylene, and further embedded in Roti-Histokitt II (Carl Roth, Karlsruhe, Germany) before coverslipping.

## Spine and Dendrite quantification

For each bird, z-stacks were obtained from three brain sections of nucleus HVC and two sections of nucleus RA with a Nikon Eclipse Ti microscope (Nikon, Tokyo, Japan), equipped with a 60x oil immersion lens (CFI Plan Apo VC 60x Oil). The outline of HVC and RA in each section was delineated in ImageJ (http://rsb.info.nih.gov/ij/), and 100 µm² non-overlapping regions of interests (ROIs) were randomly placed within the boundaries of the target nucleus prior to the quantifications. Considering the relatively uniform distribution of X-projecting HVC neurons and RA-projecting HVC neurons (*Fortune and Margoliash, 1995*), this approach should result in a homogenous sampling across treatment groups.

Neuronal dendritic spine densities were estimated for each brain section by checking ROIs in random order until five ROIs were located that contained spinous dendrite segments (15 ROIs per animal for HVC and 10 ROIs for RA). The individual dendrite segments within the ROIs were traced in ImageJ to measure the length, and the number of visible spines that originated from the traced segment was manually counted. Spine densities were calculated as the number of visible spines per μm of dendrite. All visible spine types were included in the quantifications, including filopodia, long, thin, stubby, mushroom and branched spines (*Risher et al., 2014*). For branched spines, each spine head was counted separately.

To further obtain a density measure of spinous dendrites we counted the total number of spinous dendrite segments in all ROIs examined, and divided the number of observed dendrite segments with the number of ROIs, giving an estimation of the number of spinous dendrites per ROI. Neuronal dendrites were further binned into categories containing 0–0.2, 0.2–0.4, 0.4–0.6, 0.6–0.8, 0.8–1.0, 1.0–1.2, 1.2–1.4, 1.4–1.6, 1.6–1.8 spines per μm. The probability distribution of neuronal dendrite segments was then obtained by dividing the observed number of spinous dendrites per ROI in each bin with the total number of spinous dendrites per ROI.

To obtain a density measure of aspinous neurites in HVC, the number of visible spine-less neurite segments were counted in ten new non-overlapping ROIs that were randomly placed within the boundaries of HVC. We refer to these projections as neurites, because we were unable to distinguish between aspinous dendrites and axons in our tissues.

All quantifications were conducted by an experimenter that was blind to the experimental condition of the animals.

## Statistical analysis

Statistical analyses were performed in SAS 9.3 (SAS Institute, Cary, NC). No sample sizes were calculated prior to the experiments. A technical replication was defined as a repeated measurement of the same individual, and biological replications constitute the number of sampled individuals (n). Differences in spine and dendrite densities between treatment groups were tested with a one-way analysis of variance (ANOVA) with Dunnett's correction for multiple comparisons against one control value (untreated birds). Changes in hormone levels were determined using a repeated one-way ANOVA with Dunnett's correction. Differences in syllable similarity and syllable rates of individual syllable types were tested with one-way ANOVAs, and when significant, Tukey's HSD test was used to analyze differences between time points. Song pattern similarities between the 1st and 2nd hormone treatment were established by comparing the average CC from seven consecutive days of stable song during the 1st treatment with the average CC from seven consecutive days of stable song production during the 2nd treatment. Paired-samples t-tests were used to compare CC's, sequence parameters, and developmental time parameters between subsequent hormone treatments. To determine song feature deterioration we calculated the difference between the average CC from the last 7 days of stable song production during the 1st testosterone treatment and the average CC from the first 2 days of song production during the 2nd treatment. Feature recovery was determined by subtracting the average CC from the first 2 days of song production from the average CC from the last 7 days of stable song production during the 2nd testosterone treatment. One-sample t-tests were used to determine if feature deterioration and recovery significantly exceeded zero. Measurement values in the text are given as means ± SEM, unless stated otherwise. No outliers or data were excluded from analysis.

## Acknowledgements

The authors would like to thank Ingrid Schwabl, Monika Trappschuh and Wolfgang Goymann for performing and analyzing the hormone radioimmunoassays. We also thank Sébastien Derégnaucourt for providing male canary songs and Susan Urbanus and Lasse Jacobsen for constructive comments on the manuscript.

# Additional information

## Funding

| Funder | Grant reference number | Author |
|---|---|---|
| Horizon 2020 Framework Programme | 701660 | Michiel Vellema |
| National Research, Development and Innovation Office Hungary | K-115970 | Sándor Zsebők |
| Interuniversity Attraction Poles | IUAP-NIMI-P6/38 | Annemie Van der Linden |
| Max-Planck-Gesellschaft | Open-access funding | Manfred Gahr |
| National Research, Development and Innovation Office Hungary | K-129215 | Sándor Zsebők |
| National Research, Development and Innovation Office Hungary | PD-115730 | Sándor Zsebők |

The funders had no role in study design, data collection and interpretation, or the decision to submit the work for publication.

## Author contributions

Michiel Vellema, Conceptualization, Data curation, Formal analysis, Funding acquisition, Validation, Investigation, Visualization, Methodology, Writing—original draft, Project administration, Writing—review and editing; Mariana Diales Rocha, Data curation, Formal analysis, Validation, Investigation, Methodology, Writing—review and editing; Sabrina Bascones, Data curation, Formal analysis, Investigation, Writing—review and editing; Sándor Zsebők, Formal analysis, Funding acquisition, Methodology, Writing—review and editing; Jes Dreier, Stefan Leitner, Formal analysis, Investigation, Writing—review and editing; Annemie Van der Linden, Conceptualization, Resources, Supervision, Funding acquisition, Writing—review and editing; Jonathan Brewer, Resources, Data curation, Supervision, Methodology, Writing—review and editing; Manfred Gahr, Conceptualization, Resources, Data curation, Supervision, Funding acquisition, Methodology, Project administration, Writing—review and editing

## Author ORCIDs

Michiel Vellema (iD) https://orcid.org/0000-0001-7172-4776
Stefan Leitner (iD) http://orcid.org/0000-0002-9482-0362

## Ethics

Animal experimentation: Experimental procedures were conducted according to the guidelines of the Federation of European Animal Science Associations (FELASA) and approved by the Ethical Committee on animal experiments of the University of Antwerp (protocol number: 2007-14).

## Decision letter and Author response

Decision letter https://doi.org/10.7554/eLife.43194.023
Author response https://doi.org/10.7554/eLife.43194.024

# Additional files

## Supplementary files

• Transparent reporting form
DOI: https://doi.org/10.7554/eLife.43194.019

## Data availability

SQL data of extracted song features for each individual have been deposited in Dryad. Further source data files have been provided for Figure 2, Figure 2—figure supplement 1, Figure 3, Figure 3—figure supplement 1, Figure 4, Figure 4—figure supplement 1, Figure 4—figure supplement 2, Figure 5.

The following dataset was generated:

| Author(s) | Year | Dataset title | Dataset URL | Database and Identifier |
|---|---|---|---|---|
| Vellema M, Gahr M | 2018 | Data from: Vocal motor experiences consolidate the vocal motor circuitry and accelerate future vocal skill development | https://dx.doi.org/10.5061/dryad.kb814nh | Dryad Digital Repository, 10.5061/dryad.kb814nh |

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
