## [Decision Letter]

Thank you for submitting your article "Vocal motor experiences consolidate the vocal motor circuitry and accelerate future vocal skill development" for consideration by *eLife*. Your article has been reviewed by three peer reviewers, and the evaluation has been overseen by Richard Ivry, serving as Reviewing Editor and Senior Editor. The reviewers have opted to remain anonymous.

The reviewers have discussed the reviews with one another and the Reviewing Editor has drafted this decision to help you prepare a revised submission.

Summary:

Testosterone treated female canaries are proposed as a model system to study the behavioral trajectories of seasonal song re-acquisition, taking advantage of the fact that female canaries can produce male-like song following systemic application of testosterone. The authors use this artificially-induced fast song acquisition process to study the fast re-acquisition of a previously learned skill. The authors quantified the behavioral changes during the first acquisition of a fine motor behavior, singing, to the behavioral progresses observed upon re-acquisition of the same skill after a period of absence of practice. In parallel, they investigated the changes in synaptic dendritic spine density in HVC that accompany these different phases of recurring motor development. Building on the female canaries' sensitivity to testosterone, the approach allows the experimenters to elicit on command (by adding/removing testosterone implants) the first acquisition of a song, impose a period of no practice (implant removal) and finally trigger re-acquisition of singing (new testosterone implant).

The results make a strong case that the song, upon reacquisition is similar to that acquired during the first phase, and that the behavioral course for first song acquisition is different than for re-acquisition, where only acoustic parameters linked to time (syllable rate, FM, AM, inter-syllable intervals) need to be reacquired. The stability of the song following reacquisition is particularly interesting in multiple regards: a) Unlike in male canaries (where song changes across seasons) the female song remains the same and b) The re-acquisition process of the same motor skill can be studied. The spine density results show that after the first acquisition of the song, the spine density diminished by 30% and this persisted after removal of the testosterone implant and during the absence of singing behavior. This finding is in line with the idea that synaptic pruning is responsible for improving and consolidating complex skilled movements. Although, the relationship between performance changes (from a slow first time learning process to fast relearning of a motor skill) and structural changes (pruning of spines in HVC) is purely correlational, the study represents a very creative approach to study the neuronal underpinnings of fast re-learning of already learned skilled motor behaviors.

Primary Issues for revision:

1) Since the song similarity claim is essential for the impact of this study (Figure 4), it is important to make this comparison as rigorously as possible, analyzing the similarity between the songs produced at the end phase of treatment 1 and the songs 2 produced during treatment 2. For example, it would be good to develop some sort of similarity score. Some ideas here include looking at the occurrence of same syllable sequences for a given bird compared to when comparing the productions of two birds. Or looking at the cross correlation between two conspecific testosterone treated female canaries. One important analysis is the quantification of syllable duration across treatment periods. Syllable duration did not deteriorate which strongly hints towards the re-acquisition of the same song. This finding might provide some evidence that single syllables are consolidated as one unit.

2) One interesting aspect of song, especially in the canary (as compared to the zebra finch), is that the transitions from one syllable to another obey certain probabilistic rules that are most likely dictated by HVC. While this is the case in male natural singing behavior, are females treated with testosterone singing a variable song with some transition rules or a very stereotyped song with strict rules? Were the transition rules affected by the testosterone treatment?

3) When looking at spine density only, can you differentiate dendritic pruning from apoptosis of part of the cell population? If the total number of cells is not stable, then the number of synapses should also change? Can the authors control for the cell density in HVC (or use the constant total number of neurites as a proxy)?

4) Need for more quantification of the synaptic changes undergone by the network (both HVC and RA) in all of the groups (including the +T2 group) will increase significantly the interest of this study for a broad readership. As it is, the timing coincidence between synaptic pruning in HVC and behavioral changes is an interesting piece of evidence for circuit reorganization but its link to behavior is relatively week. The extension of the quantification (what is happening in RA, analysis of the T_2_+ birds) would help make the case that synaptic changes may be linked to the behavior shown here. While we're probably not expecting a second pruning event, if the hypothesis is that the first pruning would be some sort of a motor memory trace of the song acquired under T+1, it could be worth checking that the re-acquisition of the song does not engage a second event of pruning.

5) With the experimental design, the decrease in spine number could also be explained by the application of testosterone alone and, thus, being independent of motor skill consolidation. How can the authors disentangle the effect of the treatment versus relearning?

6) The link between synaptic pruning in HVC and the fast re-learning process is a bit loose. Structural changes in HVC are linked to vocal learning in songbirds, but the specific pruning in HVC following song acquisition shown here has not been described previously. It is however difficult to parse out a possible effect of the different auditory experience (due to auditory feedback during singing) and the specific contribution of sensorimotor learning. Moreover, changes in HVC happening during the initial acquisition period may not be linked to the capacity of birds to reacquire the task rapidly. Indeed, some song parameters that are rapidly re-acquired are thought to depend more on the neural activity of nucleus RA (frequency and amplitude modulation) as they do not correlate with HVC activity in finches. These issues should be considered in revision.

7) Please provide more information about the auditory experience of the birds used for histology.

Secondary comments:

Some ideas for future work that came up in the discussion and you may wish to comment on:

1) This model could be used to investigate the difference between the motor memory trace of the song and the auditory memory trace, if the females were to match a template they heard before. Did the author control for the songs that the females might have heard from birth until Testosterone treatment? Are there any indications of song copying behavior?

2) Could it be possible to manipulate the dendritic growth during the resting period (implant removal) and see whether the re-acquisition of the song follows, after a second implant, a first-time learning trajectory or, even, the same first-time learning trajectory?

In males songbirds, it is known that with seasonal testosterone increases, there is an increase in the number of HVC cells (for instance Brenowitz and Larson, 2015). Is there anything known for females treated with testosterone?

Figure 2F: example spectrographs -> spectrograms

Figure 3—figure supplement 1: "could not be calculated for T_2_+ in F, I, as we observed no developmental increase of this song feature" -> "of these song features".

Figure 5: Dendrite ratio is unclear here. Do you mean probability or percentage?

Even looking at the method in the third paragraph of the subsection “Spine and dendrite quantification”, I don't quite get this ratio. By likelihood I assume you mean probability. Are these probabilities calculated per ROI over the total number of segments in a given ROI and then averaged over ROIs? Are you effectively calculating the probability of a density of spine given the probability of observing a spinous dendrite in an ROI?

Subsection “Song analysis”, fourth paragraph: Time = duration or delay to stabilization?

Subsection “Song analysis”, last paragraph: spectrographs -> spectrograms.

The title should be shorter and more concise.

Abstract: Skilled motor behaviors are not necessarily a response to another action but can be executed independent of prior inputs.

Introduction, first paragraph: What is meant by 'manipulative skills'? Retrieving the memory of food cashes is not a motor skill.

Introduction, second paragraph: Is it re-acquisition of the same song or learning new songs? Can the correlation of structural changes within HVC be explained if the birds learn a new song? Why should existing structures facilitate learning a new motor skill? What is the impact of the development of physical characteristics, constraints due to external factors e.g. muscle constraints or internal factors e.g. genetic background? This topic is discussed at the end but it might be useful to mention it during the Introduction.

Discussion, first paragraph: Overview of motor savings should be mentioned before own findings.

Discussion, second paragraph: This statement is out of context because the question was not addressed within this study. The authors should discuss the known difference of acquisition periods between fine versus gross motor skills.

Discussion, sixth paragraph: The authors argue that vocal maintenance can be explained by 'structural consolidation of […] X-projecting HVC neurons'. In zebra finches it has been shown that the deletion of X-projecting HVC neurons does not lead to an impairment of stable song production. How does this finding fit in?

Figure 1A It should be indicated that the depicted spectrogram is the spectral derivative or a regular spectrogram presentation can be used.

Figure 1B Recording of the same bird or different birds?

Figure 1D When were the songs recorded? Days post treatment? Is it true that testosterone treated females sing faster (more syllables per sec) compared to male canaries? Can the authors correlate singing behavior (amount of singing/ quality of song) with testosterone levels?

Supplementary Figure 1 is missing?

Figure 2: Can the authors compare the song learning of male canaries with the testosterone treated females in respect to syllable production rate? In the second testosterone treatment phase the three syllable speeds reemerge instantaneously. How do the authors explain that the birds in T_2_+ sing the slow and the medium syllables reliably more often than during phase T1? Doesn't this argue against the idea that the animals have to re-acquire the motor skill but remain proficient without practice?

Figure 2B: What is the indication of a negative cross correlation of syllable rate during the first 40 days post treatment? Is this example bird an outlier? As far as I understand the data analysis the cross-correlation was calculated based on the histogram pattern. How can a negative correlation value occur? General request: For reproducibility and the understanding of the study it is crucial that the reader can retrace whether the effect depends on single individuals or if it is an effect that is reflected in the entire group. Mean value presentations should be substituted by individual data points.

Figure 5A: When where the example photomicrographs taken (time point in relation to testosterone implant)? Figure 5B: What is the meaning of 'a' and 'b'?

Are the behavioral results displayed as mean over subjects or mean over all song syllable analyzed? This is important to make this point explicit since one bird may contribute disproportionally to the entire data set in term of number of syllables produced.

Subsection “Singing-related pruning of neuronal dendritic spines”, first paragraph: this sentence summarizing the current knowledge of neuronal processes linked to song learning is a bit minimal. It would be important to highlight the known contributions of HVC and RA to song control and the evidence for plasticity at various sites of the song control network.

In Figure 4C, it is unclear whether the correlation coefficients for the T_2_+ columns are computed among syllables produced in the T_2_+ condition or between syllables from the T_1_+ and T_2_+ conditions. As written, I understand that it's among T_2_+ syllables. If true, this measures the similarity of syllables within a condition, while the sentence 'both temporal and spectral features demonstrated a strong similarity in their distribution patterns between subsequent testosterone treatments' rather suggests that the authors want to point to the similarity between T_1_+ and T_2_+. Either the analysis or the phrasing should be changed.

Figure 5C-E: I do not understand the statement made concerning this figure: 'Instead of an equal reduction in spines across all neurite types, we observed a shift in the dendrite ratio from densely packed dendrites towards less dense dendrites in the testosterone treatment and removal groups (Figure 5C-E)". The distribution of spine density across neurites appears shifted to the left, reflecting a global decrease in spine density in my opinion. If not, the authors should explain why.

Statistical comments/presentation of data:

It would be nice if the authors could show the distribution of raw data along with mean and SEM in their graphs instead of bar plots of mean and SEM.

More information should be given as of how the acoustic parameters were measured. Did the authors used Sound Analysis Pro for instance for these as well? Does the FM correspond to the CV of the fundamental frequency? etc.

[Editors' note: further revisions were requested prior to acceptance, as described below.]

Thank you for resubmitting your work entitled "Accelerated redevelopment of vocal skills is preceded by lasting reorganization of the song motor circuitry" for further consideration at *eLife*. Your revised article has been favorably evaluated by Richard Ivry (Senior and Reviewing Editor) and three reviewers.

We are quite pleased with the revision. There are some issues that we would like you to address, but as you will see, these are really suggestions about how to improve the presentation of the material and conceptual issues you may wish to raise in the Discussion. I do not see any of these as really "major".

Conceptual/clarification issues for revision:

1) Clarify how FM can be lost but bandwidth maintained at the beginning of T2. Seems that these two parameters should go hand in hand given the pure tone types of syllables. Adding supplementary figures that show the distributions for every parameter akin to Figure 2A and along with some spectrograms examples of these deteriorations or preservations would help better picture the phenomenon.

2) What is mean by declarative features? Declarative memory? Is the idea that this is about the auditory template/memory of (a) tutor song(s).

3) You responded that the spine density was calculated '….of randomly selected spinous dendrite segments [of excitatory neurons in HVC…'. The study by Kornfeld et al., 2017, demonstrates that spine density of HVC-X projecting neurons is significantly larger than HVC-RA projecting neurons. How did you achieve a random selection of spinous segments in order to ensure that HVC-X and HVC-RA projecting neurons were sampled uniformly? Would it be possible to only sample HVC-X or HVC-RA projecting neurons?

4) Since this study is focused on the changes of spines of excitatory neurons, you may wish to address the importance of testosterone and/or vocal practice on changes in inhibition in the Discussion. It has been shown that inhibition, especially in HVC, is one of the crucial factors that is guiding the song learning process in zebra finches. Could a robust inhibitory network be the mechanism that allows seasonal birds (or in this case testosterone treated females) to re-learn their song more efficiently?

5) The second paragraph of the Discussion section is unclear. As defined (difference between max and min peak frequency) and given the sound produced by this species and papers by Beckers, Suthers and ten Cate, 2003, and Riede et al., 2006 that showed that several bird species filter out harmonics with their vocal tract to only leave the fundamental, bandwidth likely corresponds to the range of fundamental frequencies the bird is engaging for each syllable type. So bandwidth would be determined by the contraction frequency of the syrinx. Mean frequency and entropy, however, could be determined by the shape of the vocal tract more than by the syrinx contraction frequency. Could there be a statement here about aspects of the songs that are linked to the vocal tract modulation and that are preserved vs parameters linked to the speed of contraction of certain syrinx muscles that need to be reacquired? And/or add references and discussion about vocal production mechanisms to push such an argument.

6) In the third paragraph of the Discussion, does "learned" mean "memorized" here or did females produce these type of syllables (even rarely) before the first testosterone treatment? This may be an important distinction between the two mechanisms of song acquisitions. If females never sung such song syllables before the first treatment, females can potentially only rely on an auditory memory template of songs they heard (not demonstrated here but that could be another future investigation) or driven by some internal auditory template of what a male canary song should sound like (the implication/necessity of the auditory feedback for song production in females under T should by the way be tested by deafening or something similar) while during the second treatment, females can rely on a motor memory/trace of what they learned to produce during the first phase.

---

## [Author Response]

Primary Issues for revision:1) Since the song similarity claim is essential for the impact of this study (Figure 4), it is important to make this comparison as rigorously as possible, analyzing the similarity between the songs produced at the end phase of treatment 1 and the songs 2 produced during treatment 2. For example, it would be good to develop some sort of similarity score. Some ideas here include looking at the occurrence of same syllable sequences for a given bird compared to when comparing the productions of two birds. Or looking at the cross correlation between two conspecific testosterone treated female canaries. One important analysis is the quantification of syllable duration across treatment periods. Syllable duration did not deteriorate which strongly hints towards the re-acquisition of the same song. This finding might provide some evidence that single syllables are consolidated as one unit.

One of the strengths of our proposed model system is the recurrence of highly similar song patterns, and we agree that it is important to rigorously demonstrate this similarity. The data shown in Figures 2, 3 and 4 all include correlations between the songs produced at the end phase of treatment 1 and the songs produced during treatment 2, providing a measure of song similarity between the two treatments. To support this data we have included two additional song analyses: (1) For each bird we calculated the Euclidean distance between pairs of syllables of the same type from the consecutive treatment periods, providing us with a measure of similarity between groups of syllables. This analysis shows that syllables from the second treatment phase are structurally not different from syllables from the first treatment (Figure 4—figure supplement 1; subsection “Song similarity during subsequent testosterone treatments”, second paragraph of the Results and subsection “Song analysis”, sixth paragraph of the Materials and methods). (2) We calculated the probability distribution of syllable phrase transitions, which provides information on how syllable phrases are ordered in the song. This analysis shows that song sequence parameters are highly similar between the two treatment periods (Figure 4—figure supplement 2; subsection “Song similarity during subsequent testosterone treatments”, third paragraph).

2) One interesting aspect of song, especially in the canary (as compared to the zebra finch), is that the transitions from one syllable to another obey certain probabilistic rules that are most likely dictated by HVC. While this is the case in male natural singing behavior, are females treated with testosterone singing a variable song with some transition rules or a very stereotyped song with strict rules? Were the transition rules affected by the testosterone treatment?

Testosterone-treated female canaries generally sang a variable song with some transition rules, similar to what has been shown for male canaries. Songs are not random, but also not very stereotyped. Not surprisingly, birds with larger syllable repertoires tended to have more variation in how syllable phrases were sequenced than birds with smaller repertoires. The number and probability of phrase transitions in the songs did not differ between treatment 1 and treatment 2, and we did not observe significant changes in song sequence parameters, suggesting that the rules governing these phrase transitions did not change between the two treatment periods. This data has been included in Figure 4—figure supplement 2, subsection “Song similarity during subsequent testosterone treatments, third paragraph,” of the Results and subsection “Song similarity during subsequent testosterone treatments”, last paragraph, of the Materials and methods.

3) When looking at spine density only, can you differentiate dendritic pruning from apoptosis of part of the cell population? If the total number of cells is not stable, then the number of synapses should also change? Can the authors control for the cell density in HVC (or use the constant total number of neurites as a proxy)?

A distinction has to be made here between how we calculated spine densities and how we calculated dendrite densities. (1) Spine densities (Figure 5B and C) were calculated as the number of spines per µm of dendrite for a number of randomly selected spinous dendrite segments, and not per volume unit of HVC and RA. By itself this analysis is independent of how many neurons or dendrites were present in the target brain region, and apoptosis or other changes in cell density therefore do not affect this calculation. (2) Dendrite densities are represented as the probability to detect a spinous dendrite within the volume of HVC (Figure 5D-F), and are therefore susceptible to changes in cell density. We did not observe a significant difference in the density of the total number of spiny dendrites in HVC between treatments however, indicating that the shift in the probability distribution shown in Figure 5D-F is not likely to be caused by a reduction in neuronal dendrites. Taking both pieces of evidence into account, these findings suggest a reduction of synapses in HVC through selective pruning of dendritic spines, without reducing the total number of dendrites in HVC. This point has been clarified in the second paragraph of the Results subsection “Singing-related pruning of neuronal dendritic spines”.

4) Need for more quantification of the synaptic changes undergone by the network (both HVC and RA) in all of the groups (including the +T2 group) will increase significantly the interest of this study for a broad readership. As it is, the timing coincidence between synaptic pruning in HVC and behavioral changes is an interesting piece of evidence for circuit reorganization but its link to behavior is relatively week. The extension of the quantification (what is happening in RA, analysis of the T_2_+ birds) would help make the case that synaptic changes may be linked to the behavior shown here. While we're probably not expecting a second pruning event, if the hypothesis is that the first pruning would be some sort of a motor memory trace of the song acquired under T+1, it could be worth checking that the re-acquisition of the song does not engage a second event of pruning.

We have included an additional analysis of spine densities in nucleus RA demonstrating that this area does not undergo specific spine pruning during the acquisition and loss of song in adult female canaries, and suggests that the delayed maturation of the motor circuitry may be specific for HVC. This data has been included in Figure 5C and subsection “Singing-related pruning of neuronal dendritic spines”, second paragraph. Unfortunately we have no brain material available to test if a second pruning event would take place during a second testosterone treatment. The experiment was specifically designed to test if we could find an anatomical ‘memory’ trace that would last through a period in which the birds did not sing and could explain the rapid re-acquisition of a previously learned song pattern. Considering the extent of pruning that took place in HVC during song acquisition it seems highly unlikely that a pruning event of such magnitude would repeat itself during a second testosterone treatment in females, or occur on an annual basis in male canaries.

5) With the experimental design, the decrease in spine number could also be explained by the application of testosterone alone and, thus, being independent of motor skill consolidation. How can the authors disentangle the effect of the treatment versus relearning?

The reviewer raises an important question on the mechanisms behind selective stabilization and elimination of dendritic spines, mechanisms that are largely unknown. In songbirds testosterone levels are inseparably linked to the development and consolidation of vocal performance, both in our study as well as in nature. From our study we cannot determine if vocal practice, testosterone levels, or a combination of both cause the observed spine pruning in HVC. Singing prevention in zebra finches delays the developmental pruning of dendritic spines in RA at least to some extent (Hayase S, et al., 2018), suggesting that vocal practice induces spine pruning in this brain region. The authors did not control for testosterone levels however, and mechanisms of spine pruning in RA may differ from those in HVC (Figure 5B and C). Independent of the role that testosterone and vocal practice play in consolidating the song system, the reduction in dendritic spines was not reversed during a long period in which both testosterone levels were low and vocal practice was absent, providing a possible anatomical basis for the rapid recurrence of song patterns. We discuss this issue in detail in the ninth and tenth paragraphs of the Discussion.

6) The link between synaptic pruning in HVC and the fast re-learning process is a bit loose. Structural changes in HVC are linked to vocal learning in songbirds, but the specific pruning in HVC following song acquisition shown here has not been described previously. It is however difficult to parse out a possible effect of the different auditory experience (due to auditory feedback during singing) and the specific contribution of sensorimotor learning. Moreover, changes in HVC happening during the initial acquisition period may not be linked to the capacity of birds to reacquire the task rapidly. Indeed, some song parameters that are rapidly re-acquired are thought to depend more on the neural activity of nucleus RA (frequency and amplitude modulation) as they do not correlate with HVC activity in finches. These issues should be considered in revision.

In our study we associate lasting structural changes in HVC with the development and re-acquisition of vocal motor performance. Our study does not directly address sensorimotor integration, and it is currently unclear if adult canaries can learn song from an external song template. Auditory deprivation during juvenile development leads to aberrant songs in canaries (e.g. Lehongre, C. et al., 2006 Anim. Behav. 72: 1319-1327) indicating the importance of audition for vocal learning. Testosterone-treated adult female canaries on the other hand develop species-typical songs even when acoustically isolated during the time they start singing for the first time, suggesting that adult song development does not necessarily depend on the same processes as juvenile song learning. In our study we cannot directly distinguish between the effects of vocal motor practice and auditory feedback on motor circuit consolidation. We consider these issues together with the potential roles that HVC and RA may play in the formation of motor memories in the fifth paragraph of the Discussion.

7) Please provide more information about the auditory experience of the birds used for histology.

All birds used for histology were raised and kept in group aviaries together with other male and female canaries throughout the experiment to assure similar social and auditory conditions. This information has been added the first paragraph of the subsection “Specimen preparation”.

Secondary comments:Some ideas for future work that came up in the discussion and you may wish to comment on:1) This model could be used to investigate the difference between the motor memory trace of the song and the auditory memory trace, if the females were to match a template they heard before. Did the author control for the songs that the females might have heard from birth until Testosterone treatment? Are there any indications of song copying behavior?

Therefore we did not control for the auditory background of the birds during the first year of their lives prior to the experiment. We agree that our model would be well suited to investigate differences between procedural and declarative memory formation. We would speculate that declarative song memory, or the so called song template, would primarily be formed during juvenile development and is dependent on auditory input during that time, since our birds were able to develop species-typical songs in adulthood without being exposed to external auditory input. Unpublished song playback trials have not revealed clear song copying in adult female canaries, but it is currently unknown if adult female canary songs are composed of song patterns that the birds have memorized during juvenile development. This is partly discussed in the third and fifth paragraphs of the Discussion.

2) Could it be possible to manipulate the dendritic growth during the resting period (implant removal) and see whether the re-acquisition of the song follows, after a second implant, a first-time learning trajectory or, even, the same first-time learning trajectory?

We agree that this would be a very interesting approach to demonstrate causality between dendritic spine pruning and the observed rapid re-acquisition of previously learned song patterns. The mechanisms behind spine pruning or their possible regrowth are not clear, and to our knowledge no molecular techniques currently exist to induce specific regrowth of previously-pruned dendritic spines.

In males songbirds, it is known that with seasonal testosterone increases, there is an increase in the number of HVC cells (for instance Brenowitz and Larson, 2015). Is there anything known for females treated with testosterone?

Testosterone-treated female canaries demonstrate a similar increase in HVC size and HVC neuron number (e.g. Hartog et al., 2009), without changing neuron density. We included a discussion on this topic in the sixth paragraph of the Discussion.

Figure 2F: example spectrographs -> spectrograms

Corrected.

Figure 3—figure supplement 1: "could not be calculated for T_2_+ in F, I, as we observed no developmental increase of this song feature" -> "of these song features"

Corrected.

Figure 5: Dendrite ratio is unclear here. Do you mean probability or percentage?Even looking at the method in the third paragraph of the subsection “Spine and dendrite quantification”, I don't quite get this ratio. By likelihood I assume you mean probability. Are these probabilities calculated per ROI over the total number of segments in a given ROI and then averaged over ROIs? Are you effectively calculating the probability of a density of spine given the probability of observing a spinous dendrite in an ROI?

Figure 5D-F shows the probability distribution of dendrites with different spine densities. We calculated the probability distribution by dividing the number of spinous dendrites in each bin with the total number of spinous dendrites. This has been clarified in Figure 5 and in the third paragraph of the Materials and methods subsection “Spine and dendrite quantification”.

Subsection “Song analysis”, fourth paragraph: Time = duration or delay to stabilization?

‘Time’ has been replaced by ‘duration’.

Subsection “Song analysis”, last paragraph: spectrographs -> spectrograms.

Corrected.

The title should be shorter and more concise.

To convey a more concise message we changed the title to: ‘Accelerated redevelopment of vocal skills is preceded by lasting reorganization of the song motor circuitry’.

Abstract: Skilled motor behaviors are not necessarily a response to another action but can be executed independent of prior inputs.

‘Response’ has been replaced by ‘behaviors’.

Introduction, first paragraph: What is meant by 'manipulative skills'? Retrieving the memory of food cashes is not a motor skill.

With manipulative skills we mean the motor movements from e.g. paws to retrieve hidden food items. To avoid confusion with spatial memory retrieval we replaced the example with a less ambiguous example of motor skill acquisition. ‘Capuchin monkeys need to acquire manipulative foraging skills to retrieve difficult-to-access, seasonally available food items’.

Introduction, second paragraph: Is it re-acquisition of the same song or learning new songs? Can the correlation of structural changes within HVC be explained if the birds learn a new song? Why should existing structures facilitate learning a new motor skill? What is the impact of the development of physical characteristics, constraints due to external factors e.g. muscle constraints or internal factors e.g. genetic background? This topic is discussed at the end but it might be useful to mention it during the Introduction.

This sentence refers globally to seasonal song re-acquisition. Different bird species may adopt different strategies, but for adult male canaries it is factually not known if they relearn new songs each year, or if they re-develop song patterns that they have learned during juvenile development. Increasing evidence, including this study, suggests that in canaries the latter is the case. As such we correlate the structural changes that we observe in HVC with the re-acquisition of a previously learned vocal pattern. Since this is an extensive topic that depends for a part on the findings from this study, we prefer to discuss this issue in detail in the third paragraph of the Discussion.

Discussion, first paragraph: Overview of motor savings should be mentioned before own findings.

We moved the examples describing motor savings up to the first paragraph of the Discussion.

Discussion, second paragraph: This statement is out of context because the question was not addressed within this study. The authors should discuss the known difference of acquisition periods between fine versus gross motor skills.

We restructured the Discussion considerably. Because songbirds are often used as a model for sensorimotor learning, we thought it useful to specifically mention that it is not known if adult canaries can learn through sensorimotor integration. This point is more generally discussed in the third paragraph of the Discussion.

Discussion, sixth paragraph: The authors argue that vocal maintenance can be explained by 'structural consolidation of […] X-projecting HVC neurons'. In zebra finches it has been shown that the deletion of X-projecting HVC neurons does not lead to an impairment of stable song production. How does this finding fit in?

We have included a discussion on this finding in context to similar studies that argue for a role of the cortico-basal ganglia circuit in the long-term maintenance of song (Discussion, eighth paragraph).

Figure 1A It should be indicated that the depicted spectrogram is the spectral derivative or a regular spectrogram presentation can be used.

We have included this information in the legends of Figure 1, 2 and 4.

Figure 1B Recording of the same bird or different birds?

The shown spectrograms are recordings from the same individual at different times during development. This information has been clarified in the figure legend of Figure 1.

Figure 1D When were the songs recorded? Days post treatment? Is it true that testosterone treated females sing faster (more syllables per sec) compared to male canaries? Can the authors correlate singing behavior (amount of singing/ quality of song) with testosterone levels?

Recording dates have been included in the legend of Figure 1. We have no evidence that testosterone treated females can sing faster than males. The range of syllable rates observed in this study falls within the natural range of syllable rates that has been observed in male canaries living in the wild. This information has been included in the second paragraph of the subsection “Re-application of testosterone triggers an accelerated re-acquisition of song performance”. We do not have a sufficient spread of testosterone levels within and between birds to provide a meaningful quantitative correlation of song behavior with testosterone levels, except for the fact that birds do not sing when testosterone levels are very low, but sing on average ~30.000 syllables per day when testosterone levels are high.

Supplementary Figure 1 is missing?

Figure supplements are provided for main Figures 2, 3, and 4.

Figure 2: Can the authors compare the song learning of male canaries with the testosterone treated females in respect to syllable production rate?

Syllable rates of stable song from testosterone treated females fell within the natural range of syllable rates that have been observed in male canaries living in the wild (Leitner, S. et al., 2001). This information has been included in the second paragraph of the subsection “Re-application of testosterone triggers an accelerated re-acquisition of song performance”. We don’t have longitudinal song data from sufficient juvenile males to quantitatively compare syllable rate development in males and females.

In the second testosterone treatment phase the three syllable speeds reemerge instantaneously. How do the authors explain that the birds in T_2_+ sing the slow and the medium syllables reliably more often than during phase T1? Doesn't this argue against the idea that the animals have to re-acquire the motor skill but remain proficient without practice?

The three syllable rates did not reemerge instantaneously but took 26 days to stabilize in the individual shown in Figure 2A. This can be seen by the red fit curves in the histogram that do not start at their final value for any of the three distinguishable syllable rates. In addition, Figure 2B shows for the same individual how much the SR pattern during treatment 2 correlates with the SR pattern from treatment 1 illustrating a clear phase in which the SR pattern from treatment 2 redevelops and increases in similarity in respect to the SR pattern from treatment 1. On average across all birds SR patterns stabilized in 22 days during treatment 2, which was significantly faster than during treatment 1 (160 days). This information is quantitatively shown for all animals in Figure 2C-E and described in the third paragraph of the subsection “Re-application of testosterone triggers an accelerated re-acquisition of song performance”.

Figure 2B: What is the indication of a negative cross correlation of syllable rate during the first 40 days post treatment? Is this example bird an outlier? As far as I understand the data analysis the cross-correlation was calculated based on the histogram pattern. How can a negative correlation value occur? General request: For reproducibility and the understanding of the study it is crucial that the reader can retrace whether the effect depends on single individuals or if it is an effect that is reflected in the entire group. Mean value presentations should be substituted by individual data points.

We calculated Pearson’s product-moment correlation coefficient based on the histogram pattern to determine similarity between treatment periods. Pearson’s CC has a value between -1 and 1, where 1 describes two patterns that are identical, and -1 describes two patterns that are completely opposite to each other (and thus are negatively correlated). Values around zero indicate no correlation. This is described in detail in the fourth paragraph of the Materials and methods subsection “Song analysis”. We have included raw data point for all behavioral bar graphs, and further provide this information in the accompanying source data for each figure.

Figure 5A: When where the example photomicrographs taken (time point in relation to testosterone implant)? Figure 5B: What is the meaning of 'a' and 'b'?

The time of sacrifice of the birds shown in Figure 5A has been included in the figure legend and is further described in the first paragraph of the Materials and methods subsection “Specimen preparation”. ‘a’ and ‘b’ in Figure 5B illustrate if a significant difference was observed between treatment groups. This is indicated in the legend of Figure 5.

Are the behavioral results displayed as mean over subjects or mean over all song syllable analyzed? This is important to make this point explicit since one bird may contribute disproportionally to the entire data set in term of number of syllables produced.

All means in the text and figures describe the mean over subjects, with the exception of Figure 2F which illustrates mean values of syllable rates for one syllable type in one individual. This has been clarified in all figure legends and is also described in the Materials and methods subsection “Statistical analysis”. In addition we have included raw data points in bar graphs to illustrate the spread between individuals.

Subsection “Singing-related pruning of neuronal dendritic spines”, first paragraph: this sentence summarizing the current knowledge of neuronal processes linked to song learning is a bit minimal. It would be important to highlight the known contributions of HVC and RA to song control and the evidence for plasticity at various sites of the song control network.

This sentence refers to motor learning and memory in general with the purpose of providing a rationale of why we decided to look specifically at spine pruning in relation to the observed vocal motor savings. We agree that the known contributions of HVC and RA to song control and evidence for plasticity in the song system are important items to discuss, but introducing these topics in this part of the Results section would overshadow the results. We have included an extensive discussion of the roles of HVC and RA during song control and testosterone-related plasticity in the song system in the sixth and seventh paragraphs of the Discussion.

In Figure 4C, it is unclear whether the correlation coefficients for the T_2_+ columns are computed among syllables produced in the T_2_+ condition or between syllables from the T_1_+ and T_2_+ conditions. As written, I understand that it's among T_2_+ syllables. If true, this measures the similarity of syllables within a condition, while the sentence 'both temporal and spectral features demonstrated a strong similarity in their distribution patterns between subsequent testosterone treatments' rather suggests that the authors want to point to the similarity between T_1_+ and T_2_+. Either the analysis or the phrasing should be changed.

The CC’s presented in the T_2_+ columns are computed between syllables from the T_1_+ and T_2_+ conditions. The aim of the Figure 4C is to illustrate that the correlations between T_1_+ and T_2_+ patterns are no different from the correlations between T_1_+ and T_1_+ patterns, and thus the same song patterns are produced during T_1_+ and T_2_+. We realize that the figure labels for these comparisons may have caused confusion. To clarify the comparisons we changed the color legends in the figure with more intuitive labels (*T_1_xT_1_* vs. *T_1_xT_2_*). In addition we clarified the wording of the figure legend of Figure 4C and the first paragraph of the Results subsection “Song similarity during subsequent testosterone treatments”.

Figure 5C-E: I do not understand the statement made concerning this figure: “Instead of an equal reduction in spines across all neurite types, we observed a shift in the dendrite ratio from densely packed dendrites towards less dense dendrites in the testosterone treatment and removal groups (Figure 5C-E)". The distribution of spine density across neurites appears shifted to the left, reflecting a global decrease in spine density in my opinion. If not, the authors should explain why.

We fully agree with the reviewer that the change in the dendrite distribution reflects a global decrease in spine density. With ‘unequal reduction’ we attempted to refer to the changed shape of the distribution (which would keep the same shape if the quantified number of dendrites in each bin would be reduced by an ‘equal’ amount). Instead the distribution of dendrites shifts to the left from more densely-spined dendrites towards less densely-spined dendrites and supports our finding that spine densities reduce during the first testosterone treatment period. We have modified this sentence to make this message more clear (subsection “Singing-related pruning of neuronal dendritic spines”, second paragraph).

Statistical comments/presentation of data:It would be nice if the authors could show the distribution of raw data along with mean and SEM in their graphs instead of bar plots of mean and SEM.

We have included the raw data point in all bar graphs that represent behavioral data, and provide source data for each figure.

More information should be given as of how the acoustic parameters were measured. Did the authors used Sound Analysis Pro for instance for these as well? Does the FM correspond to the CV of the fundamental frequency? etc.

We used Sound Analysis Pro to calculate acoustic parameters for each syllable. We included this information in the second paragraph of the Materials and methods subsection “Song analysis”, and refer to the accompanying publication and online manual for the detailed mathematical descriptions of each sound feature. Sound Analysis Pro calculates FM as the angular component of squared time and frequency derivatives. Effectively FM describes the angle of frequency traces in a syllable and is expressed in degrees.

[Editors' note: further revisions were requested prior to acceptance, as described below.]

We are quite pleased with the revision. There are some issues that we would like you to address, but as you will see, these are really suggestions about how to improve the presentation of the material and conceptual issues you may wish to raise in the Discussion. I do not see any of these as really "major".Conceptual/clarification issues for revision:1) Clarify how FM can be lost but bandwidth maintained at the beginning of T2. Seems that these two parameters should go hand in hand given the pure tone types of syllables. Adding supplementary figures that show the distributions for every parameter akin to Figure 2A and along with some spectrograms examples of these deteriorations or preservations would help better picture the phenomenon.

Bandwidth (defined here as the difference between the max and min peak frequency) describes the range of peak frequencies of a syllable while FM is calculated in Sound Analysis Pro as the mean angular component of all frequency contours within a syllable (and not only of the peak frequency). The mean FM of a syllable thus increases with an increased up and down modulation within the duration of the syllable, but bandwidth does not necessarily change as long as these increased modulations in frequency occur within the original range of peak frequencies.

To further illustrate the detailed song changes, we have included 2 supplementary figures (Figure 2—figure supplement 2 and Figure 3—figure supplement 2) with example histograms of each song feature and included several spectrogram examples of song syllables from the end of T_1_+, beginning of T_2_+, and end of T_2_+.

2) What is mean by declarative features? Declarative memory? Is the idea that this is about the auditory template/memory of (a) tutor song(s).

This term is no longer used here, as we have changed the focus of this section from a discussion on motor constraints in general to the contribution of the syrinx and vocal tract in shaping and re-shaping song output (Discussion, second paragraph).

3) You responded that the spine density was calculated '….of randomly selected spinous dendrite segments [of excitatory neurons in HVC…'. The study by Kornfeld et al., 2017, demonstrates that spine density of HVC-X projecting neurons is significantly larger than HVC-RA projecting neurons. How did you achieve a random selection of spinous segments in order to ensure that HVC-X and HVC-RA projecting neurons were sampled uniformly? Would it be possible to only sample HVC-X or HVC-RA projecting neurons?

The sampling areas that we used for quantifications were spatially randomly distributed across HVC. Considering that HVC_X_ and HVC_RA_ neurons are relatively uniformly distributed throughout HVC (see e.g. Fortune and Margoliash, 1995) our sampling method likely captures equal ratios of both neuron types between the different treatment groups. This has been clarified in the third paragraph of the subsection “Spine and dendrite quantification”. The actual spine density of each segment does not influence the sampling frequency.

4) Since this study is focused on the changes of spines of excitatory neurons, you may wish to address the importance of testosterone and/or vocal practice on changes in inhibition in the Discussion. It has been shown that inhibition, especially in HVC, is one of the crucial factors that is guiding the song learning process in zebra finches. Could a robust inhibitory network be the mechanism that allows seasonal birds (or in this case testosterone treated females) to re-learn their song more efficiently?

We have included a discussion on the potential role of inhibitory activity during song acquisition and re-acquisition in the eighth paragraph of the Discussion.

5) The second paragraph of the Discussion section is unclear. As defined (difference between max and min peak frequency) and given the sound produced by this species and papers by Beckers, Suthers and ten Cate, 2003, and Riede et al., 2006, that showed that several bird species filter out harmonics with their vocal tract to only leave the fundamental, bandwidth likely corresponds to the range of fundamental frequencies the bird is engaging for each syllable type. So bandwidth would be determined by the contraction frequency of the syrinx. Mean frequency and entropy, however, could be determined by the shape of the vocal tract more than by the syrinx contraction frequency. Could there be a statement here about aspects of the songs that are linked to the vocal tract modulation and that are preserved vs parameters linked to the speed of contraction of certain syrinx muscles that need to be reacquired? And/or add references and discussion about vocal production mechanisms to push such an argument.

We have changed the focus of this section to discuss the potential role of the syrinx and vocal tract during song acquisition and re-acquisition, and included a statement on how they may relate to the preservation and/or deterioration of song features (Discussion, second paragraph).

6) In the third paragraph of the Discussion, does "learned" mean "memorized" here or did females produce these type of syllables (even rarely) before the first testosterone treatment? This may be an important distinction between the two mechanisms of song acquisitions. If females never sung such song syllables before the first treatment, females can potentially only rely on an auditory memory template of songs they heard (not demonstrated here but that could be another future investigation) or driven by some internal auditory template of what a male canary song should sound like (the implication/necessity of the auditory feedback for song production in females under T should by the way be tested by deafening or something similar) while during the second treatment, females can rely on a motor memory/trace of what they learned to produce during the first phase.

The female canaries in this study were not observed to produce any song syllables prior to the first testosterone treatment (subsection “Testosterone-induced song development in female canaries”). ‘Memorized’ may be a better description here and we have changed the term accordingly in the fourth paragraph of the Discussion.

We completely agree with the reviewer that the observed differences in song acquisition during the first and second treatments may depend on the absence (first treatment) and presence (secondtreatment) of a motor memory trace of the song. This idea is consistent with our view that large-scale synaptic pruning occurred in motor area HVC during the first testosterone treatment, a modification that may reflect the formation of such a motor memory trace. We have emphasized this idea in the aforementioned paragraph.